# Robust synchronization of coupled circadian and cell cycle oscillators in single mammalian cells

Jonathan Bieler[1,†], Rosamaria Cannavo[1,†], Kyle Gustafson[1], Cedric Gobet[1], David Gatfield[2] & Felix Naef[1,*]

## Abstract

Circadian cycles and cell cycles are two fundamental periodic processes with a period in the range of 1 day. Consequently, coupling between such cycles can lead to synchronization. Here, we estimated the mutual interactions between the two oscillators by time-lapse imaging of single mammalian NIH3T3 fibroblasts during several days. The analysis of thousands of circadian cycles in dividing cells clearly indicated that both oscillators tick in a 1:1 mode-locked state, with cell divisions occurring tightly 5 h before the peak in circadian *Rev-Erbα*-YFP reporter expression. In principle, such synchrony may be caused by either unidirectional or bidirectional coupling. While gating of cell division by the circadian cycle has been most studied, our data combined with stochastic modeling unambiguously show that the reverse coupling is predominant in NIH3T3 cells. Moreover, temperature, genetic, and pharmacological perturbations showed that the two interacting cellular oscillators adopt a synchronized state that is highly robust over a wide range of parameters. These findings have implications for circadian function in proliferative tissues, including epidermis, immune cells, and cancer.

**Keywords** cell cycle; circadian cycle; single cells; synchronization; time-lapse imaging
**Subject Categories** Quantitative Biology & Dynamical Systems; Cell Cycle
**Mol Syst Biol. (2014) 10: 739**

## Introduction

Understanding how cellular processes interact on multiple levels is of fundamental importance in systems biology. In this context, the interconnection between circadian and cell cycle oscillators presents an ideal system that can be analyzed in single prokaryotic (Yang *et al*, 2010) and eukaryotic cells (Nagoshi *et al*, 2004; Welsh *et al*, 2004). Interactions between the circadian oscillator and the cell cycle link two fundamentally recurrent cellular processes (Reddy &

O'Neill, 2010; Masri *et al*, 2013). The circadian clock is a cell-autonomous and self-sustained oscillator with a period of about 24 h and thought to function as a cellular metronome that temporally controls key aspects of cell physiology, including metabolism, redox balance, chromatin landscapes and transcriptional states, and cell signaling (Dibner *et al*, 2010; O'Neill *et al*, 2013). In growth conditions, successive divisions and progression through the cell cycle can also be considered as a periodic process. The cell cycle duration in mammalian cells typically also lasts on the order of 1 day (Hahn *et al*, 2009). An immediate theoretical consequence is that coupling between two such oscillators may lead to synchronization, which is also called mode-locking. In fact, depending on the relationships between the intrinsic periods of the oscillators and the strength of their coupling, the system may stabilize into a steady state in which the two cycles advance together, similar to a resonance phenomenon. More generally, the system may switch from asynchrony (quasi-periodicity) to synchronization characterized by a rational winding number (*p*:*q*) such that exactly *p* cycles of the first oscillator are completed while the second completes *q* cycles (Glass, 2001).

Studies in cyanobacteria (Mori *et al*, 1996; Yang *et al*, 2010), fungi (Hong *et al*, 2014), zebrafish (Tamai *et al*, 2012), and mammalian cells (Brown, 1991; Matsuo *et al*, 2003; Nagoshi *et al*, 2004; Kowalska *et al*, 2013) reported that cell cycle states fluctuate with circadian time. Notably, mitotic indices are known to exhibit clock-dependent daily variations (Brown, 1991; Bjarnason *et al*, 2001; Reddy *et al*, 2005; Masri *et al*, 2013). This has led to a model whereby the circadian clock may establish temporal windows in which certain cell cycle transitions are favored or suppressed, a phenomenon referred to as circadian gating of the cell cycle. Since this gating appears to be recurrent across evolution, it was proposed to reflect an adaptation, for example, to minimize genotoxic stress during DNA synthesis and replication by directing these events to time intervals of low solar irradiation and low metabolically generated oxidative stress (Destici *et al*, 2011). Improved understanding of conditions that synchronize cell and circadian cycles is of great interest for cancer chronotherapeutics, as it might help optimize the timing of anti-proliferative drug treatments (Levi *et al*, 2007).

Regulation of the cell cycle by the circadian clock involves both the G1/S and G2/M transitions. Seminal work in the regenerating

1  The Institute of Bioengineering, School of Life Sciences, Ecole Polytechnique Fédérale de Lausanne (EPFL), Lausanne, Switzerland
2  Center for Integrative Genomics, Génopode, University of Lausanne, Lausanne, Switzerland
   *Corresponding author. Tel: +41 21 693 1621; E-mail: felix.naef@epfl.ch
   †These authors contributed equally to this work

mouse liver suggested that WEE1 kinase, which limits the kinase activity of CDK1 and thereby prevents entry into mitosis, is controlled at the transcriptional level through BMAL1/CLOCK and shows circadian activity, thereby functioning as a clock-dependent cell cycle gate (Matsuo *et al*, 2003). In a single-cell study, we previously observed circadian gating of mitosis in dexamethasone-synchronized NIH3T3 fibroblasts, showing multiple windows permitting mitosis (Nagoshi *et al*, 2004). However, studies in Rat-1 fibroblasts (Yeom *et al*, 2010) and cancer cell lines (Pendergast *et al*, 2010) concluded that circadian gating of mitosis was absent. A recent breakthrough showed that NONO, an interaction partner of PER protein (Brown *et al*, 2005), gates S-phase to specific circadian times in primary fibroblasts (Kowalska *et al*, 2013). The consequences of these multiple interactions along the cell-division cycle were investigated with mathematical models, showing conditions under which the cell cycle can mode-lock to the circadian oscillator (Zámborszky *et al*, 2007; Gérard & Goldbeter, 2012). In addition, several core clock regulators including CRY proteins (Destici *et al*, 2011) and BMAL1 (Geyfman *et al*, 2012; Lin *et al*, 2013) have been shown to influence cell proliferation, although the directionality of the effects seems to be condition-specific.

Less is known about the reverse interaction, or how the cell cycle influences the circadian cycle. However, a signature thereof is the dependency of circadian period on the time of mitosis (Nagoshi *et al*, 2004). Since the circadian oscillator is based on transcriptional–translational feedback loops, it is plausible that alteration of transcription rates during cell cycle progression (Zopf *et al*, 2013), transcriptional shutdown during mitosis (Gottesfeld & Forbes, 1997), or the transient reduction in the concentration of circadian regulators following division may indeed shift the circadian phase (Nagoshi *et al*, 2004), a phenomenon that is further supported by modeling (Yang *et al*, 2008). In addition, the activation of cell cycle checkpoints, notably via the induction of DNA damage, produces a circadian phase advance (Oklejewicz *et al*, 2008; Gamsby *et al*, 2009), which is thought to involve the interactions of several circadian oscillator proteins with the CHK1,2 checkpoint kinases (Masri *et al*, 2013).

Even though the molecular interactions between the cell cycle and circadian clock are emerging, it is not clear under which conditions these lead to entrainment of one cycle by the other, or possibly synchronization between the two cycles in mammalian cells. Here, we performed a systematic analysis of the coupling between the cell cycle and the circadian clock using time-lapse imaging of mouse fibroblasts containing a fluorescent reporter under the control of the circadian clock. Semi-automatic single-cell segmentation, tracking of circadian rhythms in single cells, and estimation of the timing of divisions allowed us to gather sufficient statistics to quantitatively probe interdependencies of the two processes under a wide set of conditions, including several serum concentrations, different temperatures, treatment with pharmacological compounds to perturb one or both of the cycles, and shRNA-mediated knockdown of circadian regulator. We found that the two oscillators showed a clear signature of mutual synchronization, with cell divisions occurring very tightly 5 h before the peak of expression of the BMAL1/CLOCK-controlled circadian *Rev-Erbα*-YFP reporter. While coupling in either direction may cause such synchrony, mathematical modeling

of our data unambiguously showed that the influence of the cell cycle on the circadian clock dominated in NIH3T3 cells and that this interaction was highly robust across the many conditions tested.

# Results

### Circadian and cell cycle oscillators are tightly synchronized in NIH3T3 cells

A universal property of interacting oscillators is the emergence of synchronized states, also called mode-locking (Glass, 2001). Since the cell cycle duration in many mammalian cells lines is in the range of the period of the circadian oscillator (about 24 h), this leads to the possibility that the two cycles could synchronize. To quantitatively investigate this possibility in single cells, we used the well-established mouse NIH3T3 cell line as a model of the circadian oscillator, previously engineered with a destabilized and nuclear-localized YFP circadian fluorescent reporter driven by the *Rev-Erbα* promoter (Nagoshi *et al*, 2004). *Rev-Erbα* is a direct target of the circadian activator complex CLOCK/BMAL1, and is thus maximally expressed at midday, or at the circadian time (CT) CT6 in mouse liver (Preitner *et al*, 2002; Rey *et al*, 2011).

To monitor individual cells, we designed large-scale time-lapse microscopy experiments, in which we optimized imaging conditions for reliable cell segmentation and cell tracking. Quantification of the YFP signal intensity in individual cell nuclei allowed us to monitor circadian phase and cell division events, marked by a characteristic and short (30–60 min) dip in signal intensity due to breakdown of the nuclear envelope (Fig 1A and Supplementary Fig S1). Across several conditions, these experiments collectively produced over 10,000 cell traces, totaling 20,000 circadian peaks and 13,000 cell divisions (Materials and Methods and Supplementary Movie S1). We chose as our default condition to monitor the system at steady state and thus used unstimulated cells to reduce possible transient effects. Recordings were acquired for 72 h at 30-min intervals under a variety of conditions.

We first considered cells grown at 37°C at several serum concentrations (in the range of 2–13% FCS), with the initial aim to probe a range of cell cycle durations. However, while serum concentration affected the fraction of mitotic cells, it had only a small effect on cell cycle duration (defined as the intervals between successive mitoses), and it showed no effect on the circadian period (Supplementary Fig S2A, B and D). The most prominent observation was that the two oscillators showed a clear signature of synchronization such that cell divisions occurred, on average, 5 h before the peak of circadian *Rev-Erbα*-YFP reporter expression, independently of serum concentration (Supplementary Fig S2C). For simplicity, we thus combined the datasets for all serum concentrations in our first analysis (Fig 1). An important property of circadian oscillations in individual cells is their inherent stochasticity, which yields successive peak-to-peak times in *Rev-Erbα*-YFP signals (hereafter referred to as circadian intervals) varying by about 10% around their mean (Nagoshi *et al*, 2004; Welsh *et al*, 2004; Rougemont & Naef, 2007). Similarly, cell cycle entry and progression through the cell cycle phases also exhibit stochasticity (Hahn *et al*, 2009). These fluctuations are clearly apparent in the timings

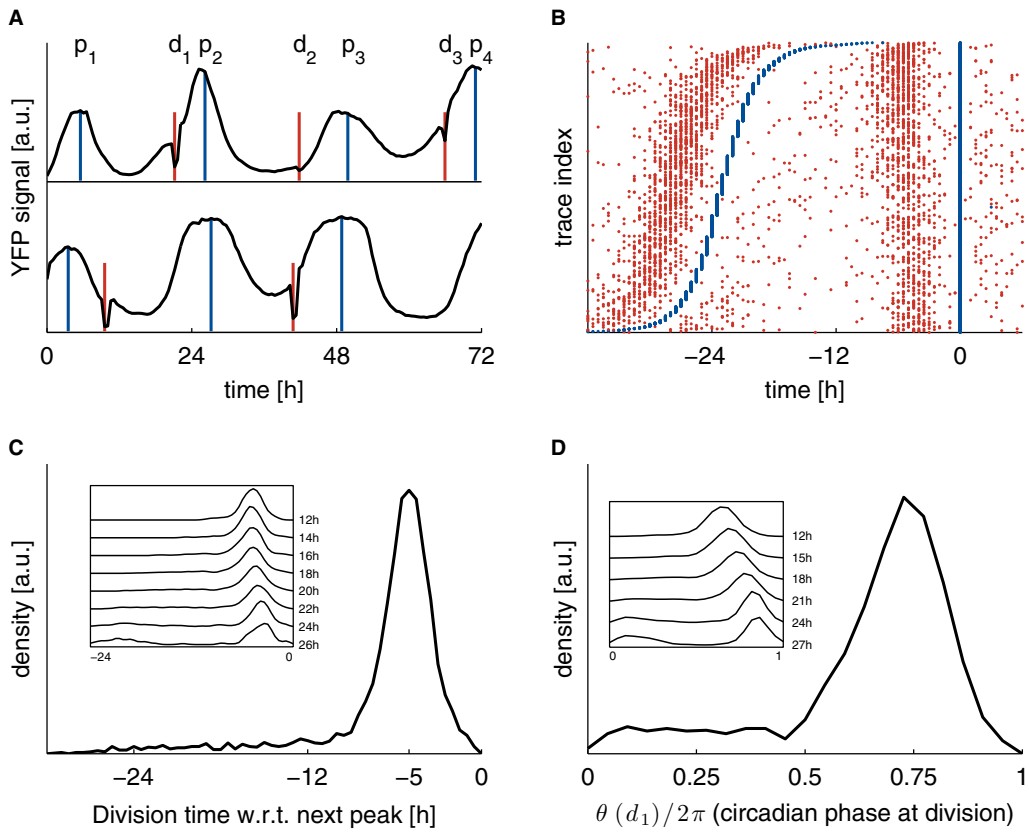

**Figure 1.  Circadian and cell cycle oscillators are tightly synchronized in NIH3T3 cells.**

A   Single-cell time traces showing the circadian YFP signal (black, identified maxima in blue denoted as p), together with cell division events (nuclear envelope breakdown, red, denoted as d). The top trace is typical and shows three divisions before the circadian peaks, the second trace shows an early first division.

B   Raster plot showing 3,160 traces (with at least two circadian peaks) aligned on the second circadian peak (blue), and sorted according to the interval between the first and second circadian peaks. Divisions (red) show a clear tendency to occur, on average, 5 h before the circadian peaks. A sparse group of early division events associated usually with longer circadian intervals is also visible.

C   Division times measured with respect to the subsequent circadian peak show a unimodal distribution centered at −5 h. Inset: longer circadian intervals correlate with mitosis occurring, on average, closer to the next peak (also visible in B).

D   Circadian phases at division (normalized division times) show a unimodal distribution. Inset: longer circadian intervals correlate with mitosis occurring at later circadian phases.

of circadian peaks and cell divisions (Fig 1A and B). It is therefore remarkable that the intervals, denoted by (d,p), between divisions (d) and following circadian *Rev-Erbα*-YFP peaks (p) show a strongly peaked and unimodal distribution centered around −5 ± 2 h (Fig 1B and C). Moreover, it was apparent that longer circadian intervals tended to include divisions that occurred closer to the next circadian peak (Fig 1B and C). The variability of (d,p) intervals was significantly smaller than that of the intervals, denoted by (p,d), from the previous peaks to the divisions (Supplementary Fig S3A). As a consequence, (d,p) intervals were also less variable compared to the circadian phases at division (division times normalized to the enclosing circadian interval, also referred to as division phases, Fig 1D). Part of this variability came from the inclusion of circadian intervals of variable duration (due to the noise), with shorter circadian intervals associated with advanced division phases, and longer circadian intervals associated with delayed division phases (Fig 1D, inset).

The significant variability in each of the cycles clearly ruled out that this tight synchrony could reflect independently running,

initially synchronized cycles. In fact, the synchrony of the circadian and cell cycles was equal for events in the first and second half of the recordings (Supplementary Fig S3B). Instead, the peaked and unimodal distribution must reflect the interaction of the two oscillators within each cell, resulting in a 1:1 mode-locked state. Furthermore, while the large majority of cells divided late in the circadian interval, a minority of cells, owing to the stochastic nature of the coupled system, divided early. This occurrence was more frequent for long circadian intervals (Fig 1A and B; see modeling below). Overall, the observed synchronization was highly robust to fluctuations. Indeed, the successive circadian intervals and cell cycle durations, measured on events $(p_1,d_1,p_2,d_2)$ or $(d_1,p_1,d_2,p_2)$, were highly correlated, although the individual circadian and cell cycle intervals varied by more than 30% (Supplementary Fig S3C, $R^2 = 0.52$, $n = 1,230$, $P < 10^{-16}$).

Thus, our data showed that circadian and cell cycles proceeded in tight synchrony in NIH3T3 cells. Translated to CT, taking the *Rev-Erbα*-YFP transcription peak as a reference (CT6), our divisions occurred near CT1, consistent with earlier observations

in mouse liver (Matsuo *et al*, 2003) and rodent epidermis (Brown, 1991). However, evidence of synchronization does not yet inform on the directionality of the interactions, as such a state could be established if either of the cycles entrained the other, or both.

### The cell cycle influences circadian phase progression

To further investigate the directionality of the interactions, we first exploited the fact that stochastic exit from the cell cycle also produces circadian intervals in which no divisions occur between two successive circadian peaks. Comparing circadian intervals with division, denoted by $(p_1,d_1,p_2)$, and those without divisions, $(p_1,p_2)$, we observed a clear shortening of the circadian interval in the presence of divisions (Fig 2A). While circadian intervals without division ($n = 2,748$) lasted $23.7 \pm 3.1$ h, as expected for free-running circadian oscillators, the intervals with one division ($n = 1,926$) lasted $21.9 \pm 3.8$ h ($P < 10^{-16}$, *t*-test), which provides an unambiguous signature that cell cycle progression influences the circadian cycle. Also, these durations were nearly identical for events from the first and second half of the recordings, thus excluding the possibility that this correlation could have originated from temporal biases in the recordings (Supplementary Fig S4). Moreover, although the majority of cell division events occurred late in the circadian interval, the duration of the circadian interval varied depending on the circadian phase at cell division (Fig 2B, an alternative representation is shown in Supplementary Fig S5A), as already reported in cells stimulated with dexamethasone (Nagoshi *et al*, 2004). Indeed, the circadian intervals were shortest (18 h on average) when mitosis occurred about halfway into the interval, while being longest (27 h) for early divisions. To investigate this further, we estimated the instantaneous circadian phase from the *Rev-Erbα*-YFP signal using a hidden Markov model (Fig 2C, Materials and Methods). This showed that compared to circadian intervals without divisions, the circadian phase progression was distorted both for cells with early and later divisions (Fig 2C and D), thus providing further evidence of a directional interaction. Indeed, cells with early divisions showed a transient slowing down of the circadian phase progression after division, while cells dividing about halfway through the circadian interval showed a speedup near and following division (Fig 2D).

This finding naturally begged the question of whether the reverse interaction, by which the circadian cycle gates the cell cycle, was evident as well. Surprisingly, the characteristics of $(d_1,p_1,d_2)$ events did not require such an interaction (compare Supplementary Fig S5A and B). Indeed, while $(p_1,p_2)$ intervals negatively correlate with $(p_2,d_1)$, $(d_1,d_2)$ positively correlate with $(p_1,d_1)$, and this positive correlation can be explained by assuming that $(d_1,d_2)$ intervals and normalized peak times $(p_1-d_1)/(d_2-d_1)$ independently vary around their means, the latter being a consequence of the entrainment of the circadian cycle by the cell cycle. No similar argument can be made to explain the negative correlation in Supplementary Fig S5A. While this suggests that no gating mechanism needs to be invoked to explain the data, further quantitative arguments will be presented in the next section. Thus, while gating of cell division by the circadian cycle in mouse cells, established in the liver (Matsuo *et al*, 2003) and in primary fibroblasts (Kowalska *et al*, 2013), has attracted the most attention, our data suggest that the influence of

the cell cycle on the circadian oscillator is predominant in NIH3T3 cells under standard culture conditions.

### A stochastic model of two coupled phase oscillators shows the dominant influence of the cell cycle on the circadian oscillator

In order to characterize the possibly reciprocal interactions more rigorously, we implemented and calibrated a mathematical model describing two interacting, noisy cycles (Equation 1 in Materials and Methods). As previously done for circadian oscillations (Rougemont & Naef, 2007; d'Eysmond *et al*, 2013) and the coupled system (Yang *et al*, 2010), we describe the two cycles by noisy phase variables (θ for the circadian and ϕ for the cell cycles) that are subject, in the absence of influences from the other oscillator, to a mean frequency modulated by prescribed noise (phase diffusion). For non-dividing cells, this model thus accounts for variable circadian intervals (for example Fig 2A, black). In addition, to encompass the three scenarios of a circadian clock gating the cell cycle, of the cell cycle influencing the circadian clock, or both, we used generic forms for the coupling function in either direction, in which each phase could slowdown and/or speedup the other phase for some combinations of phases (Supplementary Fig S6; Materials and Methods). Briefly, a function $F_1(\theta, \phi)$ represents the influence of the cell cycle phase on the circadian phase, where positive regions of $F_1$ (in yellow, Supplementary Fig S6) accelerate the circadian phase, while negative ones (in blue) slow it down. Likewise, $F_2$ represents the action of the circadian clock on the cell cycle. In order to allow for different scenarios and to keep the model complexity manageable, we parameterized $F_1$ and $F_2$ as a mixture of two weighted two-dimensional Gaussians with arbitrary means and diagonal covariance matrices (represented as ellipses in Figs 3, 4 and 6, and Supplementary Fig S6).

To fit the model to data, we computed the likelihood of the time traces by decomposing the probability of a trace as a product of causally independent factors, and approximated the probabilities of these by numerical simulations (Materials and Methods). Parameters were then estimated by maximizing the total likelihood using a genetic optimization algorithm (Hansen & Ostermeier, 2001). We used simulations to validate our fitting and assess identifiability of the parameters (Supplementary Information), which showed that the model was able to predict the directionality of the coupling and recovered the prominent features of the coupling functions.

We applied this method first to the 37°C dataset (Fig 3). The best-fit model was able to reproduce the data accurately (estimated parameters in Supplementary Tables M1–M5), as indicated by comparing data and best fit for several features: the distributions of circadian intervals, those of cell cycle durations, those of the intervals from divisions to the next circadian peaks, and those of the interval between the previous peaks and the divisions (Supplementary Fig S7). In particular, the model was able to capture the later division time observed in longer circadian intervals (Fig 3A and B). The most important features of the model are the coupling functions $F_1$ and $F_2$. Strikingly, the best-fit model predicted an acceleration of the circadian phase right around or slightly after division as the strongest interaction, when the circadian phase just passed its trough, and a weaker slowdown earlier in the circadian cycle (Fig 3C). On the contrary, the effects of the

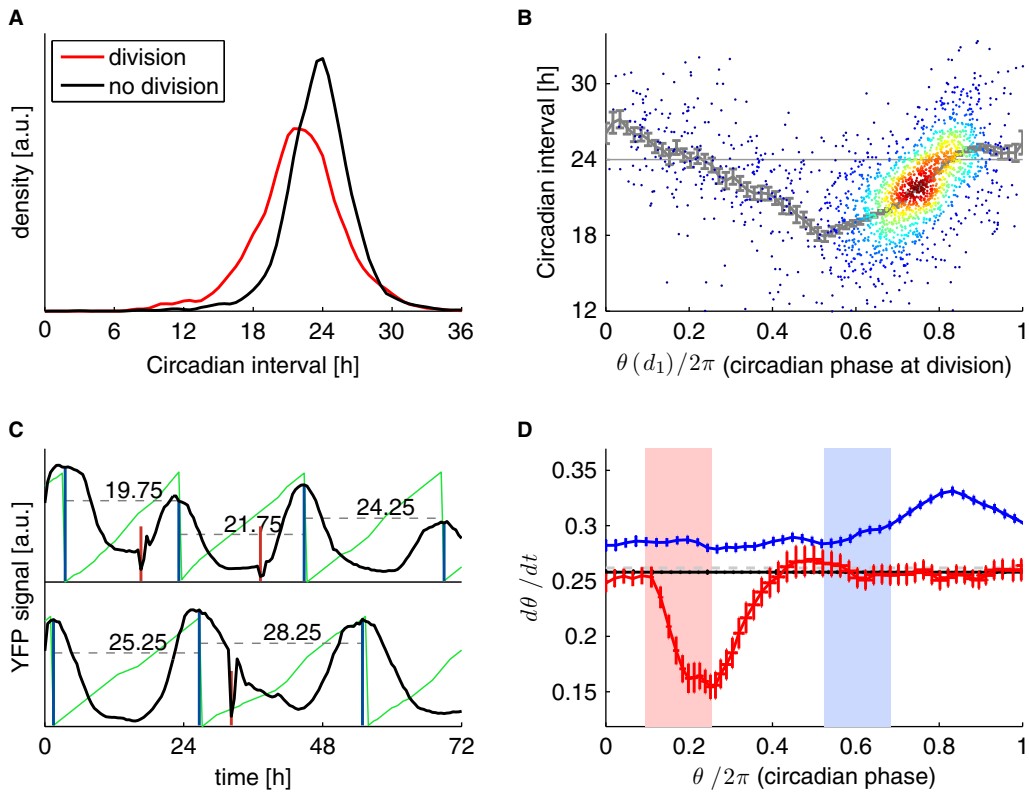

**Figure 2.  The cell cycle influences circadian phase progression.**

A  Circadian intervals with divisions ($p_1,d_1,p_2$) last 21.95 ± 3.8 h ($n$ = 1,926) and are significantly shorter ($P < 10^{-16}$, $t$-test) compared to circadian intervals with no divisions ($p_1$, $p_2$) lasting 23.7 ± 3.1 h ($n$ = 2,748).

B  Duration of circadian interval as a function of circadian phase ($\theta$) at division. The latter is estimated from interpolating between the two maxima. Running mean and standard errors are indicated in gray.

C  Estimation of the instantaneous circadian phase from the wave forms using a hidden Markov model (Supplementary Information). The instantaneous phase (thin green lines, zero phase is defined as the maximum of the waveform) shows a distortion when comparing short circadian intervals (top trace) with longer ones. Note also the slowdown of the phase progression after an early division (shown in red, bottom).

D  Instantaneous circadian phase velocity as a function of the circadian phase for intervals without divisions (black) shows that in cells with early divisions (within the pink interval, $n$ = 103), the circadian phase progression is slowed down around and after the division (red), compared to circadian intervals with no divisions ($n$ = 2,748, horizontal black line). In contrast, cells with late divisions within the light blue interval ($n$ = 234) show a globally shifted velocity and a speedup in circadian phase progression after and around the division (blue). Standard error of the mean for the instantaneous frequency at each time is indicated. For better visualization, the three velocity profiles are normalized (centered) by the nearly flat velocity profile (not shown) in division-free intervals. The gray line corresponds to $2\pi/24$.

circadian cycle on the cell cycle were much weaker. The resulting (deterministic) phase portrait shows an attracting 1:1 mode-locked state (Fig 3C), and the tendency of stochastic trajectories to cluster in the phase space according to circadian intervals (Fig 3D) explains the observed shift in division times (Fig 1D). To explore the possibility of multiple solutions among local maxima, we ran multiple optimizations with different initial conditions (parameters obtained in Supplementary Table M1). The obtained solutions indicated that the acceleration of the circadian phase close to mitosis was a robust property, while slowdown was found in some solutions and its location in the phase plane was more variable (Fig 3E). Note that these two effects were consistent with the slowing down and acceleration of circadian phase progression discussed using the instantaneous phase estimation (Fig 2C and D). Finally, while a few solutions indicated that the circadian cycle influenced cell cycle progression, the location of this gating in phase space was not consistent (Fig 3F).

As an alternative and model-independent method to deduce causal relationships among the circadian and cell cycle oscillators, we applied the Granger causality test (Granger, 1969). We used the property that nuclear size conveys information on cell cycle progression in mammalian cells (Fidorra *et al*, 1981), which we validated from time-lapse recordings in HeLa cells (Sakaue-Sawano *et al*, 2008) (Supplementary Information). We then tested whether nuclear size Granger caused the circadian *Rev-Erbα*-YFP signal, and vice versa, and found that a much larger proportion of cells (up to 60%) showed evidence ($P < 0.001$, Granger-Wald test) for a causal influence of cell cycle progression on the circadian signal, compared to the reverse interaction (< 20%) (Supplementary Fig S8). Counting only cases where the evidence was stronger in one direction compared to the other gave 55 and 12%, respectively. Altogether, our quantitative modeling of the time traces strongly suggested that the influence of the cell cycle on the circadian cycle was the dominant effect in our recordings.

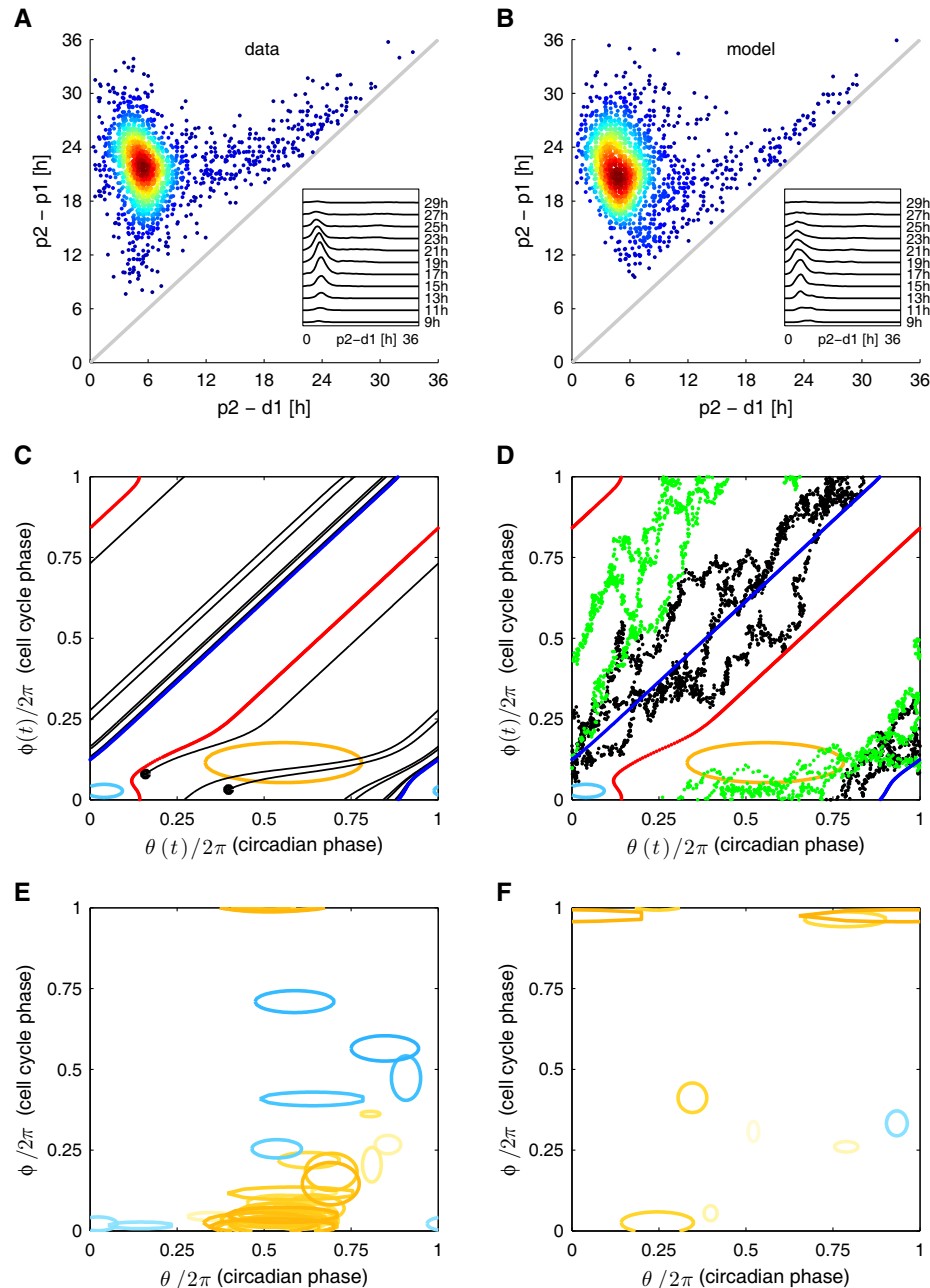

**Figure 3.  A stochastic model of two coupled-phase oscillators shows that the influence of the cell cycle on the circadian oscillator is predominant.**

A, B  Data versus model. Circadian intervals with divisions ($p_1 d_1 p_2$) as a function of the shorter subinterval ($d_1, p_2$) from the data (A) and well reproduced by the fit (B). Outliers represent a minority of cells dividing early in the circadian cycle, and the tendency of cells to divide nearer the peak for long intervals is also reproduced.

C  A generic stochastic model of two interacting phases ($\theta$: circadian phase, $\theta = 0$ is the circadian *Rev-Erbα*-YFP peak; $\phi$: cell cycle phase, $\phi = 0$ at mitosis) is fit to data, giving an estimate for the coupling functions. Phase portrait (noise terms set to zero) of the best-fit solution shows 1:1 mode locking. The blue (red) curves represent the attractor (repeller), and the black lines are representative trajectories (initial conditions shown as black dots). Regions inside the ellipses represent the influence of the cell cycle on circadian phase: significant speedup of the circadian phase occurs close to, or shortly after, cell division (yellow), while slowdown occurs for earlier circadian phases (light blue). The contours correspond to |K1*G1| or |K2*G1| = 2 [rad/h], and the reverse couplings (K3 and K4) are not shown since they are very small. Estimated parameters are given in Supplementary Tables M1–M5.

D  Stochastic simulations explain why longer circadian intervals coincide with later divisions (Figs 1D and 2B). Trajectories with long circadian intervals (black) divide late in the circadian cycle and thus tend to have short (d,p) intervals. Trajectories with short circadian intervals (green) tend to divide early in the circadian cycle and tend to have longer (d,p) intervals.

E  Coupling functions obtained describing the influence of the cell cycle phase on the circadian phase for 36 independent optimizations show consistency in the location of the acceleration of circadian phase due to the cell cycle (orange), while the slowdown is more variable and weaker in magnitude (light blue). Here 29 (7) out of 42 (30) positive (negative) Gaussians with values above 2 [rad/h] are plotted.

F  Coupling functions describing the influence of the circadian phase on the cell cycle are smaller and not consistently located in phase space. Here 7 (1) out of 38 (34) positive (negative) Gaussians with values above 2 [rad/h] are plotted.

    

## Changing temperature affects cell cycle duration and shortens circadian intervals in dividing cells, but does not disrupt synchronization

The above modeling predicts that modifying cell cycle duration should influence circadian intervals. To test this, we exploited the fact that the circadian oscillator in NIH3T3 cells is temperature compensated (Tsuchiya *et al*, 2003), while the cell cycle duration is not (Watanabe & Okada, 1967; Yeom *et al*, 2010). We thus repeated the experiment at both lower (34°C) and higher (40°C) temperatures, which indeed shifted the mean cell cycle duration by 6 h, from $24.5 \pm 4.4$ h at 34°C to $18.1 \pm 3.5$ h at 40°C (Fig 4A). As expected, the circadian intervals $(p_1,p_2)$ without divisions were effectively temperature compensated, in fact slightly overcompensated ($Q_{10} = 0.93$) but less so than reported in population experiments (Tsuchiya *et al*, 2003) (Fig 4B). But importantly, circadian intervals encompassing cell divisions gradually shortened with increasing temperature, thus confirming the prediction (Fig 4B). Interestingly, this means that temperature compensation is less effective in dividing NIH3T3 cells (here $Q_{10} = 1.36$ for intervals with divisions), and in general, temperature compensation will depend on the proliferation status of the cells. Despite these significant changes in cell cycle duration, the synchronization of the two cycles remained tight, showing a virtually indistinguishable distribution of intervals from division to the next peak (d,p) at the three temperatures (Fig 4C). Since the duration of the full intervals $(p_1,d_1,p_2)$ decreased with temperature, the divisions occurred at significantly advanced circadian phases at 40°C (Fig 4D). While we might have expected that the increased period mismatch between the circadian oscillator and the cell cycle at the highest temperature could have either disrupted synchrony or revealed mode-locking different from the 1:1 state (Glass, 2001), as in the case of cyanobacteria (Yang *et al*, 2010), we found that 1:1 locking was resilient to these changes. Moreover, the phase advance in the divisions at 40°C is consistent with the increased period mismatch, as this is a generic property of phase responses in entrained oscillators (Granada *et al*, 2013).

To assess whether our model was able to match the data at these three temperatures, we recalibrated the model to all temperatures jointly (using a single likelihood function), keeping all parameters common except for the cell cycle frequency, which was allowed to take independent values. We also reasoned that fitting more data jointly would help identify the coupling functions better. This constrained model matched the data well (Fig 4A, C and D, and Supplementary Fig S9), and the predicted shared coupling functions were qualitatively similar to the ones obtained with a single temperature (Fig 4E and F, Supplementary Tables M2–M5). The main differences were that the slowing down of circadian phase was more consistently placed toward the center of the phase plane (Fig 4E) and the weak influence of the circadian cycle on cell division seemed to be predominantly negative, as would be predicted by a gating mechanism. Therefore, our extended temperature dataset could be captured well by a model in which only the cell cycle duration was affected. Moreover, the accelerating influence of the cell cycle on the circadian phase was strong enough to maintain 1:1 mode-locking despite the period mismatch.

## Inhibition of the cell cycle lengthens circadian intervals and delays division phase

In order to complement the temperature experiments with more direct interventions on the cell cycle, we monitored cells at 37°C in the presence of inhibitors of CDK2, affecting G1/S transitions and CDK1, affecting G2/M transitions. Increasing concentration of the CDK2 inhibitor, NU-6102, did not change the duration of division-free $(p_1,p_2)$ intervals. However, it progressively increased the duration of $(p_1,d_1,p_2)$ intervals from about 22 h as in the unperturbed condition (Fig 2A) to the same duration as $(p_1,p_2)$ intervals (Fig 5A), concomitantly with an expected lengthening of the cell cycle duration (Fig 5B). Interestingly, the highest concentration (10 μM) produced significantly delayed division phases compared to the lowest concentration (1 μM) (Fig 5C). Invoking the same argument as in the 40°C temperature experiment, this delay is now consistent with a reduction of period mismatch at the higher dose. Though it was overall more difficult to record cells for 3 days under the CDK1 inhibitor, RO-3306, presumably due to higher toxicity and arrest in G2, the results were overall very similar with those of the CDK2 inhibitor, including progressive lengthening of $(p_1,d_1,p_2)$ intervals of the cell cycle duration (Fig 5D and E), and significantly phase-delayed divisions (Fig 5F). Thus, interfering with cell cycle progression at two different checkpoints confirmed that cell cycle progression has a clear and predictable influence on the duration of circadian intervals and circadian phases at division.

### *Cry2*-deficient cells with longer circadian periods do not affect the cell cycle but shift divisions

We next aimed at testing conditions in which the circadian cycle was perturbed and first opted for a genetic approach. To this end, we engineered NIH3T3-Rev-VNP1 lines stably expressing a validated shRNA targeting the *Cry2* transcript (Moffat *et al*, 2006), a condition that lengthens circadian period by a few hours (Thresher *et al*, 1998; van der Horst *et al*, 1999; Maier *et al*, 2009; Zhang *et al*, 2009). This produced the expected perturbation on the circadian oscillator (mean period of 26.3 h) but did not affect cell cycle duration (Supplementary Fig S10A, B and E), confirming that the circadian cycle did not have a strong influence on the cell cycle. However, the circadian intervals with divisions were still significantly shorter than those without divisions (Supplementary Fig S10E). In addition, the distribution of both (d,p) intervals and division phases, while still unimodal, showed a modest but significant enrichment of advanced divisions (Supplementary Fig S10C and D), again consistent with the predicted phase advance from an increased period mismatch. Thus, these *Cry2* knockdown experiments are fully consistent with the predictions of unidirectional coupling from the cell cycle onto the circadian cycle. Moreover, these data indicate that CRY2 protein is dispensable for the underlying coupling mechanism.

## Treatment with Longdaysin lengthens circadian intervals and cell cycle duration but preserves synchronization

To further probe a condition of longer circadian period, we repeated the experiments at 37°C after treating cells with Longdaysin. This

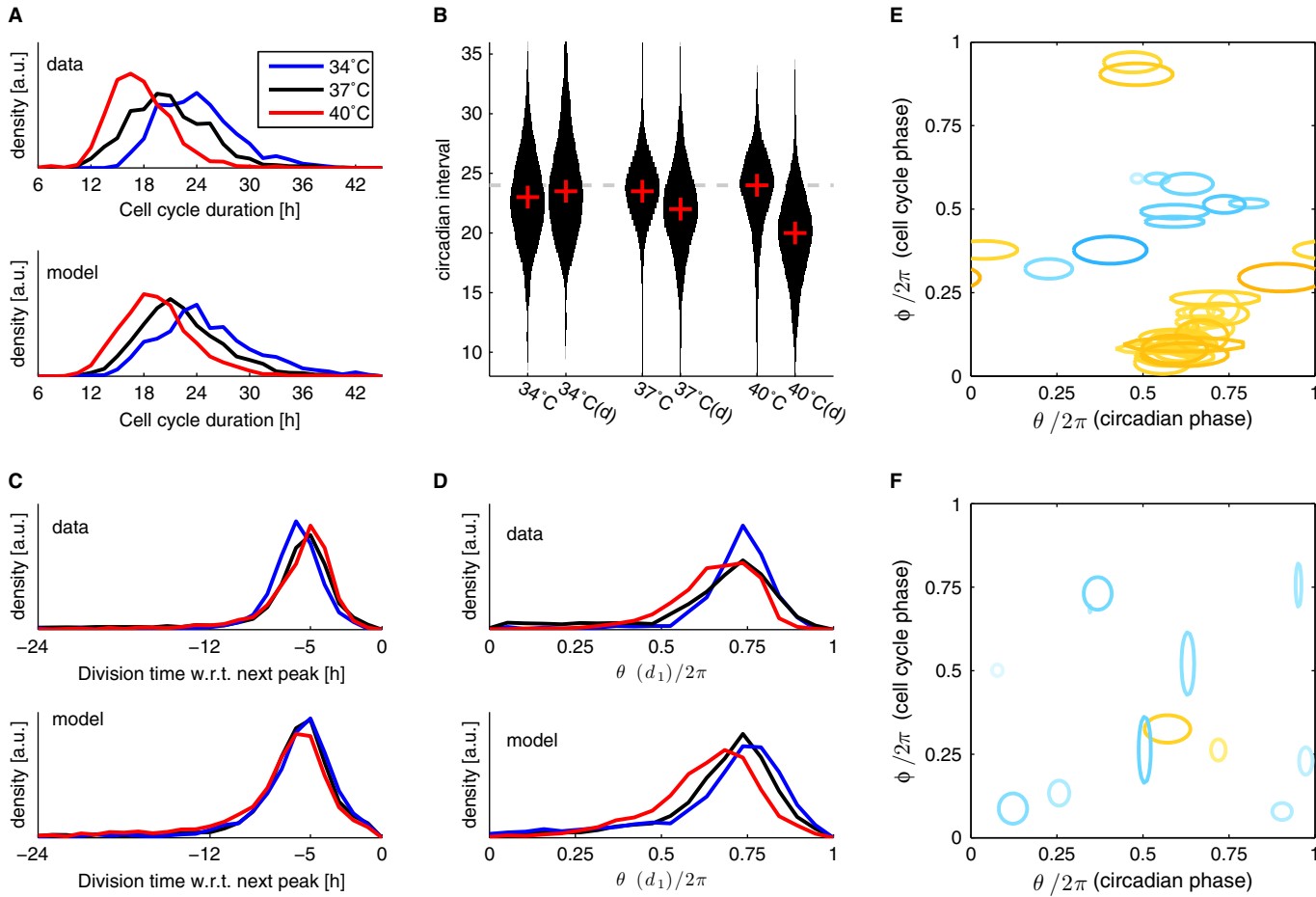

**Figure 4.  Changing temperature affects cell cycle duration and shortens circadian intervals only in dividing cells.**

A    The cell cycle duration (interval between divisions) scales with temperature.

B    Circadian intervals are temperature compensated (slight overcompensation, $Q_{10}$ = 0.9) in the absence of division (columns labeled 34, 37 and 40°C), and decrease with increasing temperature in presence of divisions (columns labeled 34, 37 and 40°C (d)). Width of the black areas indicates density of traces (histograms); the crosses indicate the median.

C    Division times with respect to the next peak are not affected by temperature: divisions occur, on average, 5 h before the circadian YFP peaks at all temperatures.

D    Circadian phases at division (normalized division times) show unimodal distributions at all temperatures. Division phases at 40°C are significantly phase advanced compared to 37°C ($P < 10^{-7}$, Kolmogorov–Smirnov test, K–S). Division phases at 34°C show a small but significant ($P < 10^{-9}$, K–S test) phase delay compared to 37°C.

E, F    Fitting data from all three temperatures together: only the intrinsic periods of the cell cycle were allowed to change, coupling parameters were shared among the three temperatures (obtained parameters are summarized in Supplementary Table M1). (E) Coupling functions obtained describing the influence of the cell cycle phase onto the circadian phase for 38 independent optimizations show consistency in the location of the acceleration of circadian phase due to the cell cycle (orange), while the slowdown (light blue) is more variable and weaker in magnitude. The contours are as in Fig 3. Here 27 (9) out of 41 (35) positive (negative) Gaussians with values above 2 [rad/h] are plotted. (F) Coupling functions describing the influence of the circadian phase onto the cell cycle are small (only 12 out of the 76 Gaussians are above threshold) and not consistently located in phase space. Here 2 (10) out of 4 (72) positive (negative) Gaussians with values above 2 [rad/h] are plotted.

Data information: The dataset included $n$ = 1,139 cell traces at 34°C, $n$ = 4,207 at 37°C, and $n$ = 1,374 at 40°C.

compound lengthens the circadian period in a dose-dependent manner through inhibition of CK1δ (Hirota *et al*, 2010), a well-known regulator of circadian period that acts by controlling the stability of PER proteins (Etchegaray *et al*, 2009). However, Long-daysin is also known to inhibit additional kinases (Hirota *et al*, 2010), of which in particular ERK2 has been noted for its role in cell cycle progression at several checkpoints in NIH3T3 (Wright *et al*, 1999) and other cells (Chambard *et al*, 2007). Our data showed that compared to control, the circadian period without divisions ($p_1$,$p_2$) progressively increased from 24 h to nearly 32 h with increasing

Longdaysin concentration (Fig 6A). As in previous conditions, ($p_1$,$d_1$,$p_2$), intervals were systematically shorter compared to ($p_1$,$p_2$) intervals at all Longdaysin concentrations. Instantaneous phase analysis showed that the circadian phase progression in treated cells was slowed down in the interval of low *Rev-Erbα*-YFP expression in a dose-dependent manner, consistent with the destabilizing effect on PER proteins of CK1δ inhibition (Etchegaray *et al*, 2009) (Supplementary Fig S11). However, the cell cycle duration also significantly increased in a dose-dependent manner. While this could in principle reflect gating of the cell cycle by the circadian clock, which our

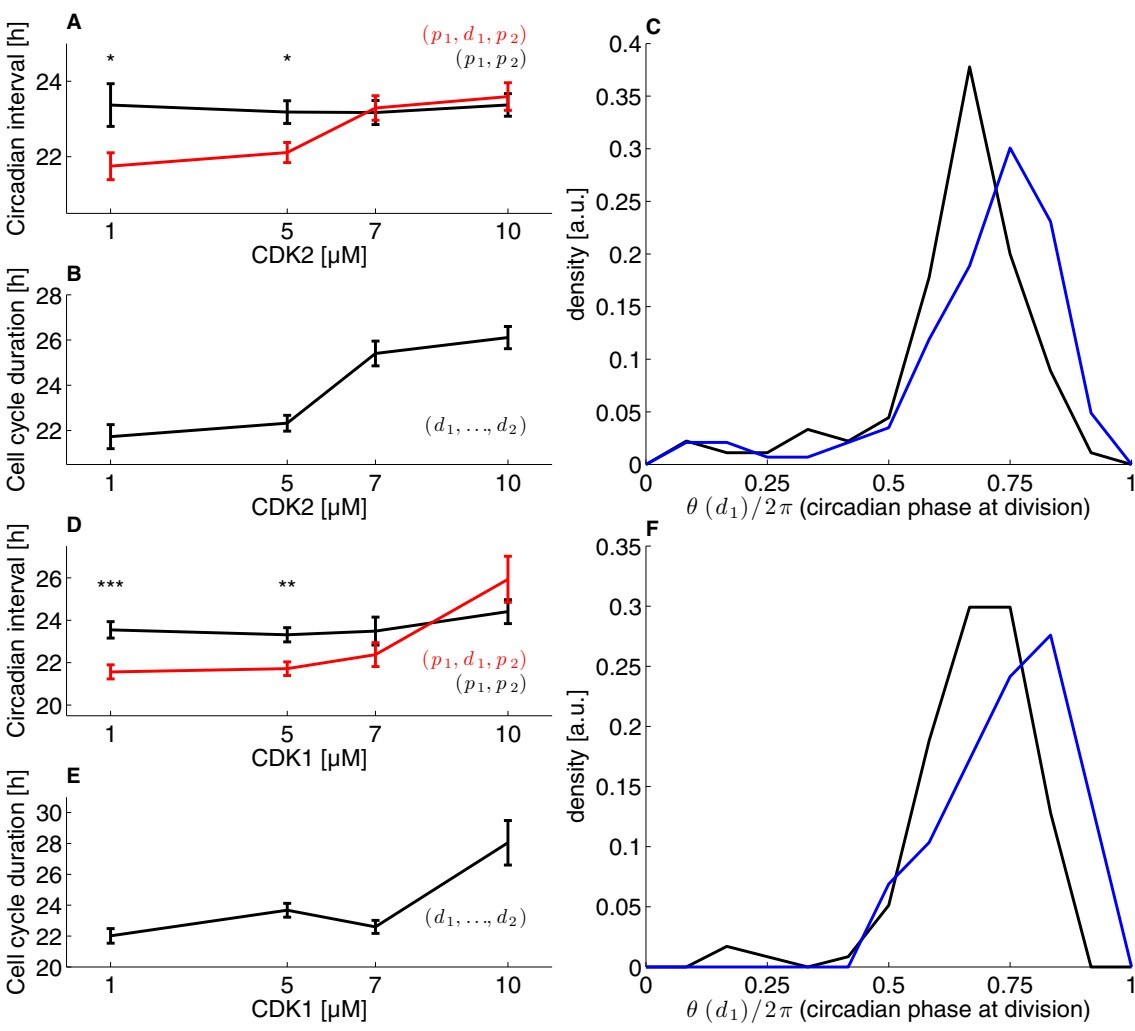

**Figure 5. Inhibition of the cell cycle lengthens circadian intervals and delays division phase.**

A   Mean circadian intervals as a function of CDK2 inhibitor concentration for intervals with division (red) and without (black) show that intervals with division lengthen as the cell cycle duration lengthen. The error bars show the standard error on the mean.

B   Mean cell cycle duration as a function of CDK2 inhibitor concentration.

C   The distribution of normalized division times (circadian phase at division) at 1 μM CDK2 inhibitor (black) and 10 μM (blue) shows a significant shift ($P < 1.2 \times 10^{-5}$, K–S test) toward later phases.

D   As in (A) for the CDK1 inhibitor.

E   As in (B) for the CDK1 inhibitor.

F   As in (C) for the CDK1 inhibitor ($P < 0.003$, K–S test).

Data information: In (A) and (D), significant difference between ($p_1$,$p_2$) and ($p_1$,$d_1$,$p_2$) intervals is indicated (*$P < 0.05$; **$P < 0.01$; ***$P < 0.001$, t-tests). The dataset included $n = 812$ cells traces for the CDK2 and $n = 711$ for the CDK1 inhibitors, nearly equally distributed across concentrations.

analysis had not revealed so far, we deemed a direct effect of Longdaysin on cell cycle progression the more likely scenario (Wright *et al*, 1999; Chambard *et al*, 2007).

Using cell counting, we indeed confirmed that Longdaysin treatment reduced the proliferation of NIH3T3-Venus cells, as well as of HeLa cells, which are devoid of circadian oscillators. These experiments thus suggested that the cell cycle period increase observed under Longdaysin treatment did not reflect gating (Supplementary Fig S12) and that Longdaysin rather induced a condition in which both the intrinsic circadian and cell cycle periods were lengthened. Remarkably, the interval lengths from divisions to circadian peaks

were sharply peaked at all Longdaysin concentrations and indistinguishable from the control condition (Fig 6B), even though the overall variability in circadian interval had nearly doubled (Fig 6A, inset). The only difference was that upon treatment, a small proportion of cells divided early in the circadian interval (Fig 6C), indicating that the 1:1 state might start to be destabilized at the highest Longdaysin concentration.

Finally, we applied our modeling to all concentrations independently. While indeed the model predicted that both circadian period and cell cycle duration were lengthened in a dose-dependent manner (Supplementary Table M1), the estimated coupling functions were

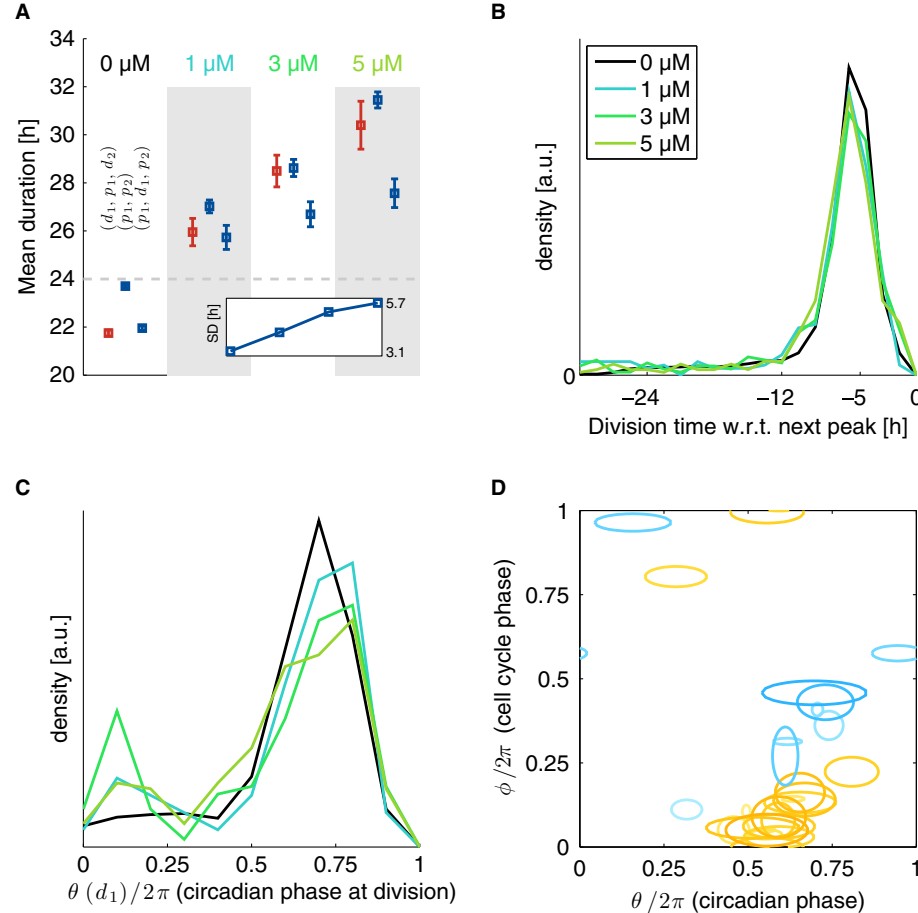

**Figure 6.   Treatment with Longdaysin lengthens circadian intervals and cell cycle durations but does not disrupt synchronization.**

A   Dose dependency of cell cycle durations ($d_1,p_1,d_2$), circadian intervals without division ($p_1,p_2$) and circadian intervals with divisions ($p_1,d_1,p_2$). Inset: dose dependency of the standard deviation (SD) of circadian intervals ($p_1,p_2$).

B   Temporal synchronization of the two cycles is equally tight at all Longdaysin concentrations and indistinguishable from the control condition.

C   Normalized division times (circadian phase at division) show that Longdaysin-treated cells have more early divisions compared to control.

D   Coupling function estimated from the stochastic model ($n$ = 31 independent optimizations) for 1,3 and 5 μM Longdaysin is similar to ones obtained in control (Fig 3). Models for all concentrations are fit independently (obtained parameters are summarized in Supplementary Table M3). Contours are as in Figs 3 and 4. Here 17 (9) out of 35 (27) positive (negative) Gaussians with values above 2 [rad/h] are plotted.

Data information: the dataset included $n$ = 1,435 cells traces nearly equally distributed across concentrations.

similar to the ones obtained in controls, again confirming the absence of clear signs of cell cycle gating by the circadian clock (Fig 6D). Furthermore, rescaling the period parameters of the models obtained for 37°C to match the observed periods in Fig 6A, while keeping other parameters (noise and coupling) fixed, was sufficient to obtain good agreement with the 5 μM Longdaysin data (mean log-likelihood of −2800 ± 50 for rescaled solutions versus −2700 ± 16 for best-fit solutions directly fitted on the Longdaysin dataset, and −4800 ± 760 for original, non-rescaled solutions). Longdaysin decreased the intrinsic frequencies of the two oscillators by a similar factor, but the coupling and noise parameters remained largely unaffected (Supplementary Table M1). After rescaling the frequencies, we obtained an effective phase model (Equation 1, Materials and Methods) in which both coupling and noise became stronger with increasing Longdayin concentrations. This might explain why synchrony was only mildly affected despite significantly increased variability of circadian intervals. Taken together, these

results are consistent with a model in which the cell cycle and the circadian clock are coupled phase oscillators, with a coupling that is predominantly from the cell cycle to the circadian clock.

## Circadian phase resetting does not influence cell divisions but transiently perturbs synchronization of circadian and cell cycles

Finally, to complement the long-lasting genetic and pharmacological perturbations of the circadian oscillator, we decided to use an approach that is much less invasive. To this end, we transiently perturbed the circadian phases using established phase resetting protocols that are based on brief treatment with dexamethasone and forskolin. While both treatments showed the expected alignment of circadian phases (Fig 7A and Supplementary Fig S13), the timing of cell divisions appeared mostly random and unaffected by the treatment (Fig 7A), indicating that this condition allowed transient uncoupling of the circadian and cell cycles. Remarkably, sorting of

the recorded cell traces according to the first division revealed that subsequent (second) circadian *Rev-Erbα*-YFP peaks tightly followed division by the usual 5 h, while cells without divisions (on the top, above the thin lines) remained aligned with the treatment (Fig 7A). We used different synchronization indices (order parameters, Supplementary Information) to quantify the differences of dexamethasone-treated and control cells as functions of time. This confirmed that circadian cycles were synchronized by the treatment, and this synchrony gradually decayed over the recording time (Fig 7B). Meanwhile, the synchronization of the cell cycle was low throughout the recordings (Fig 7C). However, the relative synchronization of the circadian and cell cycles in the treated cells, while showing a marked reduction at the beginning of the recording, eventually relaxed to identical levels as for the untreated cells after about 40 h (Fig 7D). Thus, by acutely perturbing the circadian but not the cell cycle phases, this minimally invasive perturbation of the circadian clock provided a condition in which the synchronization of the two cycles was transiently disrupted. This confirms that the circadian oscillator does not strongly influence cell division, while cell divisions determine the timing of the consequent circadian peaks.

## Discussion

### Effects of cell cycle progression on circadian oscillators

Circadian and cell cycle oscillators in individual cells and tissues provide a system in which two fundamental periodic processes may reciprocally influence each other. A number of important cell cycle regulators display 24-h rhythms in expression levels or activity that are aligned with the circadian cycle (Ueda *et al*, 2002; Miller *et al*, 2007; Gréchez-Cassiau *et al*, 2008). Molecular investigations showed that the circadian clock controls cell cycle progression both at the G1/S (Geyfman *et al*, 2012; Kowalska *et al*, 2013) and at G2/M (Matsuo *et al*, 2003; Hong *et al*, 2014) transitions, a phenomenon referred to as circadian gating of the cell cycle. On the other hand, cell cycle progression imposes rather drastic temporal changes notably on the level of transcription, which increases following replication (Zopf *et al*, 2013) and shuts down during chromosome condensation (Gottesfeld & Forbes, 1997), or via partitioning of cellular content during mitosis. Since the circadian oscillator in individual cells is highly sensitive to perturbations, as revealed through phase-shifting experiments (Nagoshi *et al*, 2004; Pulivarthy *et al*, 2007), it was natural to expect that the cell cycle could influence the circadian oscillator. It was reported previously that the time of mitosis correlates with local circadian period (Nagoshi *et al*, 2004) but also that cell proliferation reduces the coherence of circadian cycles in cell populations (O'Neill & Hastings, 2008). Here, we performed a large-scale quantitative analysis of single NIH3T3 cells carrying a fluorescent circadian phase marker under various experimental conditions, including altered growth conditions (serum and temperature), as well as genetic and pharmacological perturbations. A main result was that under steady-state free-running conditions (no entrainment), the coupled oscillators tick in a 1:1 mode-locked state that is highly resilient to perturbations, with divisions consistently occurring 5 h before the *Rev-Erbα*-YFP peak, and circadian phases at division shifting according to period mismatches in the different conditions, reminiscent of generic

properties of forced oscillators. Moreover, our modeling showed that the influence of the cell cycle on circadian phase progression quantitatively accounted for the observed mode-locking. Although this finding did not completely exclude that a circadian gating of the cell cycle occurred as well, this effect was clearly subordinate to the much stronger reciprocal interaction described above. In our data, dividing cells thus showed circadian periods that were systematically shorter by several hours, as compared to non-dividing cells (Fig 2). Conceivably, this property may also depend on the model organism or cell types. However, irrespective of possible cell type-specific variations, our findings may have important consequences for downstream circadian functions in proliferating tissues *in vivo*, and also for population measurements in cellular assays, in which circadian period is often used as a phenotype. Interestingly, a genome-wide siRNA screen in U2OS cells identified cell cycle regulators as an enriched functional category affecting circadian period (Zhang *et al*, 2009).

### Possible mechanisms mediating influence of the cell cycle on circadian phase

Cell cycle progression could influence the circadian oscillator by a number of plausible mechanisms, but for the moment we can only speculate why cell divisions lead to shortened circadian intervals. Most divisions occur at the equivalent of CT1, toward the end of the low *Rev-Erb*-YFP expression phase, when the *Rev-Erbα* promoter, activated by the BMAL1/CLOCK complex, is still in a repressed state due to nuclear CRY1 proteins bound to BMAL1/CLOCK on the DNA (Stratmann *et al*, 2010; Ye *et al*, 2011). It is thus conceivable that mitosis, by diluting (Nagoshi *et al*, 2004) or relocating CRY proteins, contributes to derepressing the *Rev-Erbα* promoter more rapidly, such that cells dividing in a CRY-repressed state would be able to initiate a new round of BMAL1/CLOCK activity more rapidly than cells that did not divide. Consistent with this scenario, our modeling found that the acceleration of the circadian phase predominantly took place just after the division (Figs 3E, 4E and 5D). Of note, due to potential inaccuracies in the instantaneous phase estimates, it is not entirely excluded that the acceleration of the circadian phase could be due to an earlier event in the cell cycle, such as during late G2, where transcription rates are higher due to double DNA content (Zopf *et al*, 2013). Concerning molecular players involved, since *Cry2*-depleted cells only showed a modest but predictable tendency toward advanced division phases, though statistically significant, we conclude that CRY2 is not a key player in mediating this coupling. Moreover, as suggested (Nagoshi *et al*, 2004), the slowdown of the circadian phase progression following early divisions (Fig 2D) could also be explained by the dilution argument, since the accumulation of the state variables PER and CRY (Travnickova-Bendova *et al*, 2002), then in their production phase, would be delayed.

### Dynamics of two coupled oscillators

Coupled oscillators are not only of great biological interest, but also very interesting from a dynamical systems standpoint. The noise-free (deterministic) dynamical behavior originating from two coupled phase variables representing the state of each oscillator is strongly constrained (since two trajectories cannot cross). Solutions therefore show either irregular (quasiperiodic) behavior

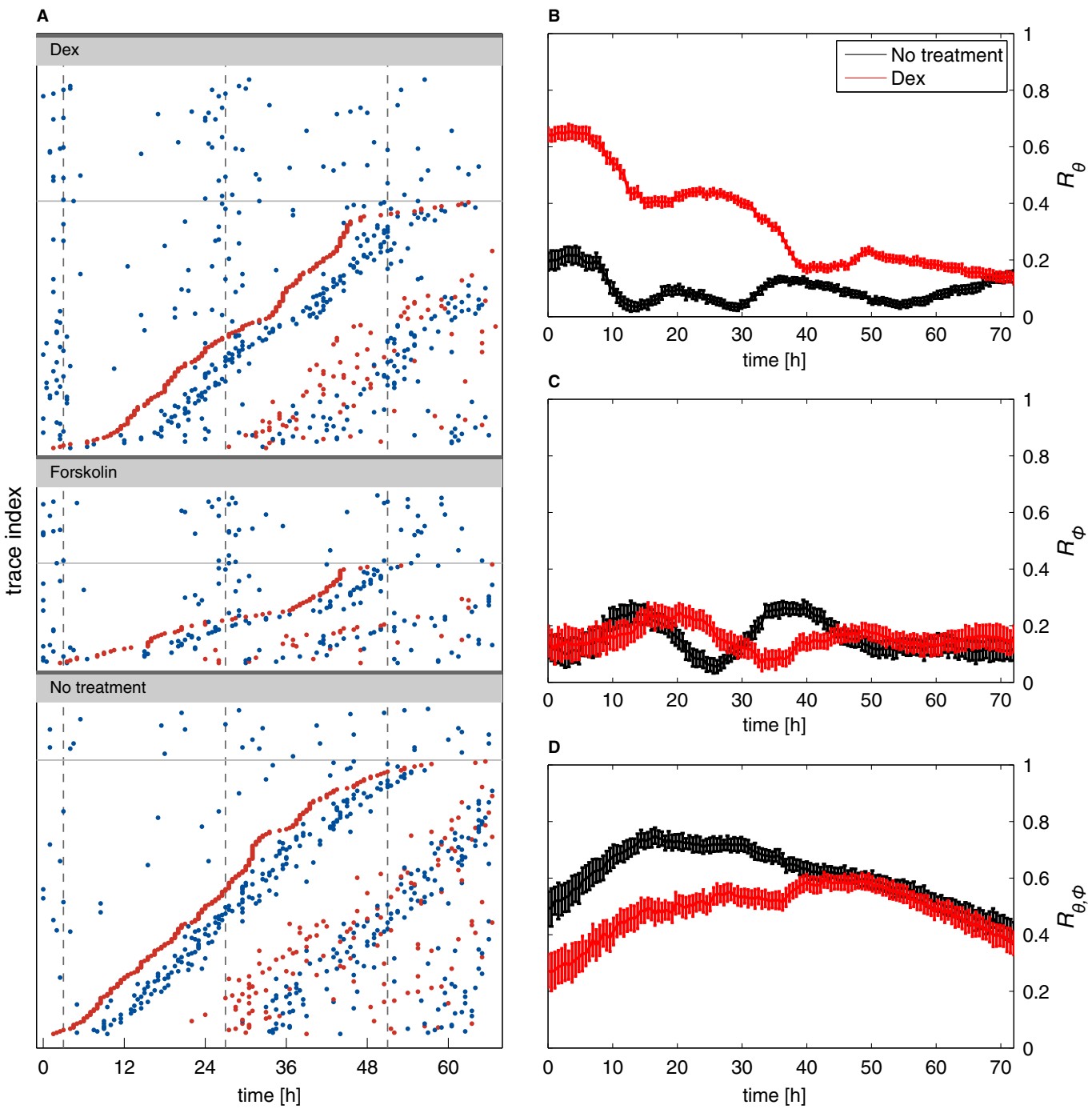

**Figure 7. Circadian phase resetting does not influence the cell cycle and transiently perturbs synchronization of circadian and cell cycles.**

A    Raster plots (each line is a cell trace) for cells treated with dexamethasone (Dex), forskolin, and untreated controls. Circadian peaks are in blue and division in red. Traces without division are in the upper parts of the panels above the thin lines. For cells with divisions, traces are sorted from bottom to top according to the time of the first division. This shows that cell divisions occur uniformly and are not affected by the phase resetting and that the second circadian peaks follow division after both dexamethasone and forskolin.

B–D  Synchronization indices over time in dexamethasone-treated cells (red) and controls (black). A value of zero for an index coincides with fully random phases while a value of 1 describes perfect synchronization. The circadian synchronization index $R_\theta$ (B) is initially much higher in dex-treated cells, as expected. Synchrony rapidly decays due to divisions (as visible in A, non-dividing cells clearly stay more synchronized). The cell cycle synchronization index (C) $R\phi$ is low throughout the recordings, indicating that dexamethasone treatment, and thus circadian phase synchronization, does not synchronize the cell cycle (also visible in A since the first divisions do not line up vertically). The synchronization index $R_{\theta,\phi}$ (D) measuring synchronization of the circadian and cell cycles indicates that dexamethasone treatment transiently reduces synchrony of the two cycles. The initial increase ($t < 15$ h) in both conditions mostly reflects larger uncertainties in the estimated phases for early times (Supplementary Information). Error bars indicate standard deviations.

corresponding to an unsynchronized state or the two phases proceed in synchrony, such that the system exhibits mode-locking. Mode-locked states are characterized by a winding number *p:q*, specifying that *p* cell cycles complete during *q* circadian cycles (Glass, 2001). Changing parameters such as the individual frequencies of the cycles or the coupling functions can drive the systems from one state to another, resulting in qualitatively distinct outcomes. For examples, two cell divisions may occur every circadian cycle (2:1) instead of one (1:1) as described in cyanobacteria (Yang *et al*, 2010). It was suggested that the multimodal distribution of division times found in NIH3T3 cells synchronized by dexamethasone (Nagoshi *et al*, 2004) may be explained by more complex (with higher *p* and *q* integers) mode-locked solutions (Zámborszky *et al*, 2007). Since molecular oscillators are subject to noise, the deterministic scenario is blurred; nevertheless, qualitative differences reminiscent of the different synchronization states remain. The observed stochastic 1:1 mode-locked state still leads to a unimodal distribution of division times, while reduced synchrony generates a significantly broader distribution and other (*p:q*) states produce multimodal distributions (Yang *et al*, 2010). The fact that, by and large, we did not observe such states even under strongly perturbed conditions, for example in the Longdaysin-treated cells, indicates that this synchrony is highly robust over a range of conditions, presumably because it confers selective advantage.

While we have addressed the problem of possible couplings in rather general terms using our inference method (Fig 3), the net dynamical mechanism leading to synchronization is relatively simple: since the cell cycle duration was mostly shorter than the circadian period in the conditions probed, synchrony resulted from the transient acceleration of the circadian phase around mitosis, leading to a stably attracting synchronized state (the attractor) in the coupled system. In fact, our data (in particular Figs 2B, D and 7) point to a scenario in which cell divisions, when occurring after a critical circadian phase (which we can tentatively assign to the nuclear entry of the PER and CRY repressors), act as a strong resetting of the circadian cycle (via derepression of the *Rev-Erbα* promoter). This then produces the tight 5-h delay of the *Rev-Erbα* peak and explains the positive slope of circadian intervals versus division phase for late dividing cells (Fig 2B). As already mentioned, this scenario would also explain why circadian intervals are lengthened when divisions occur early (Fig 2B), since a dilution of the repressors in their accumulation phase would then delay reaching of the critical phase. Note that this stochastic effect is added on top of the deterministic shifting of the peak in division phases as a function of period mismatch, observed in the temperature, the shCry2, and cell cycle inhibition experiments.

While we predominantly detected signature of the influence of the cell cycle on circadian phase progression across all conditions, we note that the Granger causality test detected at most 12% of cells that favored the reverse interaction of the circadian cycle onto the cell cycle (Supplementary Fig S8). There are several reasons why our experiments might not reveal clearer evidence for circadian gating of the cell cycle. This could originate as an experimental limitation since the circadian and cell cycle phases were observed only at certain snapshots. However, simulations suggested that this is not a severe limitation since noise actually renders the coupling functions identifiable to a reasonable extent, provided that the regions of interactions are located in a region of phase space that is explored

by the noisy dynamics under the conditions probed (Supplementary Information). Also, it is possible that biologically possible interactions are in effect inactive in certain conditions. This would typically be the case when the attractor does not intersect the regions in phase space where gating is effective. Finally, it is quite possible that gating is simply not strongly active in NIH3T3 cells, as suggested for other cell types including cancer cells (Pendergast *et al*, 2010; Yeom *et al*, 2010).

## Circadian oscillator and cell cycle in cyanobacteria

In cyanobacteria, it was reported in population studies (Mori *et al*, 1996) and single cells (Yang *et al*, 2010) that the circadian cycle can gate cell division. Time-lapse microscopy combined with mathematical modeling was thus able to show that the cell cycle is synchronized by the circadian clock, and that increased rates of cell division engender a system transition from a 1:1 to a 2:1 state in which the cells divide twice every circadian cycle (Yang *et al*, 2010). However, the reverse interaction appears to be absent, at least it does not affect the high accuracy (24-h periods) or precision (very low period dispersion) of the circadian phase (Mihalcescu *et al*, 2004). Given the significant perturbations faced by cycling cells, for example changes in cell size, doubling of DNA content, partitioning of cellular components at cell division, it is remarkable that the cyanobacterial clock circuit can buffer such nuisances.

## Relevance for circadian rhythms in proliferating mammalian cells and tissues

While most adult tissues such as the liver or the brain show little or no cell division, the interaction described is particularly relevant as recent reports indicate that the circadian clock exerts important timing functions in proliferating tissues such as epidermis (Janich *et al*, 2011, 2013; Geyfman *et al*, 2012), hair follicles (Plikus *et al*, 2013), intestinal epithelium (Mukherji *et al*, 2013), or immune cells (Cermakian *et al*, 2013; Scheiermann *et al*, 2013). The importance of 24-h timing across these systems suggests that these cell types may have found solutions to escape the period alterations induced by the cell cycle, or alternatively that systemic signals with 24-h periodicity may override or even entrain the occurring shortened periods of the cell-autonomous oscillators. This would then lead to the interesting possibility that proliferating cells within a tissue (or proliferating tissues as a whole) might show slight phase advances compared to the non-proliferating ones. An obvious next step relevant in the context of cancer chronotherapeutics (Levi *et al*, 2007) would be to extend our approach to cancer tissues, starting with the human osteosarcoma U2OS cell line, a widely used circadian model (Maier *et al*, 2009; Zhang *et al*, 2009).

In conclusion, our study sheds quantitative light on a hitherto understudied aspect of the coupled circadian and cell cycles in mammalian cells, namely that of the influence of the cell cycle on the circadian phase dynamics. While the gating of cell cycle progression by the circadian cycle has attracted most attention, we showed here that in NIH3T3 cells grown under standard conditions, the cell cycle has a dominant influence on the circadian cycle, leading to exquisitely robust synchronization of the two cycles. This possibility has important implications for chronobiology in proliferating tissues.

# Materials and Methods

### Cell culture

NIH3T3-Rev-VNP-1 cells (abbreviated NIH3T3-Venus), shScramble-NIH3T3-Rev-VNP-1 cells, shCry2-NIH3T3-Rev-VNP-1 cells, and HeLa cells were maintained in DMEM supplemented with 10% FCS and 1% PSG antibiotics. For time-lapse microscopy of fluorescent cells, the medium was replaced by phenol red-free DMEM. Unless indicated, recording conditions were at 10% FCS. When probing different FCS concentration, NIH3T3-Rev-VNP-1 cells were switched to the new concentration 1 day before starting the recordings. Where indicated, NIH3T3-Rev-VNP-1 cells were incubated with 1, 3, and 5 μM Longdaysin (Sigma-Aldrich) or 0.1% DMSO in 10% FCS phenol red-free DMEM few hours before starting the recording.

Phase resetting of the circadian cycle was performed with either 30 min 100 nM Dexamethasone-shock (Sigma-Aldrich) or by treatment with 10 μM forskolin (Biotrend). Perturbation of cell cycle progression was performed with the use of the CDK1 inhibitor RO-3306 (Sigma-Aldrich) or the CDK1/2 inhibitor II NU-6102 (Calbiochem) at the concentration of 1, 5, 7, and 10 μM.

### Fluorescence time-lapse microscopy

Cells were plated in 12-well glass bottom dishes (MatTek's Glass Bottom Culture Dishes, P12GC-1.5-14-C). The dishes were placed on a motorized stage in a 37°C chamber equilibrated with humidified air containing 5% $CO_2$ throughout the microscopy. For the temperature experiments, temperature in the chamber was modified to either 34 or 40°C, and dishes were incubated at the respective temperatures for 4 h before starting recordings. Time-lapse microscopy was performed at the EPFL imaging facility (BIOP) with an Olympus Cell Xcellence microscope using a 20× objective. The cells were illuminated (excitation at 505 nm) for 20, 40, and 60 ms every 30 min for 72 h. Time-lapse movies were captured with the use of a YFP filter set and an Andor Ixon3 camera. Images from three to four fields per well were acquired using Olympus Xcellence software.

### Cell tracking

Individual nuclei from fluorescence images were automatically segmented using a custom method (Supplementary Information, Section I) and tracked in time using a standard algorithm (Jaqaman *et al*, 2008). The timing of circadian *Rev-Erbα*-YFP peaks was automatically detected from the single-cell circadian signal while the division times were detected by using both the tracking data and the fluorescence signal. Each segmented image was manually validated and corrected, and likewise for each circadian peak and division (Supplementary Information, Section I).

### Plasmids, lentiviral production, and viral transduction

Lentiviral shRNAs in vector backbone pLKO.1(Moffat *et al*, 2006) were Scramble shRNA (addgene #1864; DNA barcode CCTAAGGT TAAGTCGCCCTCG), Cry2-targeting shRNA (Sigma-Aldrich, clone TRCN0000194121; DNA barcode GCTCAACATTGAACGAATGAA). Lentiviral particles were produced in HEK293T cells using envelope vector pMD2.G and packaging plasmid psPAX2 as previously

described (Salmon & Trono, 2007). NIH3T3-Rev-VNP1 cells were transduced with viral particle-containing supernatants according to standard procedures, and transduced cells were selected on 5 mg/ml puromycin.

### Proliferation assay

Proliferation assays were performed by counting cells using the automated cell counter Luna (Logos biosystems). HeLa or NIH3T3-Rev-VNP-1 cells were seeded in triplicate for each condition in 12-well plates and counted after 48 h for both 0.1% DMSO and 5 μM Longdaysin. Cells were trypsinized, spun down and resuspended in DMEM diluted with Trypan blue stain 0.2% (Logos Biosystem). For each biological replicate, 4–8 counts were performed.

### Instantaneous estimation of circadian phase

We inferred the circadian phase from the fluorescent *Rev-Erbα*-YFP signal using a hidden Markov model (HMM). The model contains two hidden states: the circadian phase and the signal amplitude. As in our stochastic phase model, the phase variable follows a Brownian motion with drift. The amplitude variable, necessary to account for amplitude variations in the data, is modeled as an Ornstein–Uhlenbeck process. The circadian phase is related to the data through a sinusoidal waveform. Finally, the most likely temporal sequence of phases and amplitudes was computed for each trace using the Viterbi algorithm (Supplementary Information, Section III).

### Stochastic phase model

The two cycles are modeled by noisy phase oscillators. We use θ to denote the circadian phase and ϕ for the cell cycle phase. θ = 0 corresponds to a *Rev-Erbα*-YFP peak and ϕ = 0 to a mitosis. The stochastic differential equations for the generic coupled model read:

$$
\begin{aligned}
d\theta_t &= 2\pi/T_1\, dt + F_1(\theta_t, \varphi_t)\, dt + \sigma_1\, dW_t \\
d\varphi_t &= 2\pi/T_2\, dt + F_2(\theta_t, \varphi_t)\, dt + \sigma_2\, dY_t
\end{aligned}
\tag{1}
$$

In the absence of interaction between the two cycles, the phases follow a Brownian motion with drift, with intrinsic periods $T_1$ and $T_2$ and phase diffusion coefficients $\sigma_1$ and $\sigma_2$. $dW_t$ and $dY_t$ are independent Wiener processes. The interaction between the two cycles is captured by the two functions, $F_1$ and $F_2$. $F_1$ represents the influence of the cell cycle onto the circadian clock, where positive regions of $F_1$ accelerate the circadian phase, while negative ones slow it down. Likewise, $F_2$ represents the action of the circadian clock on the cell cycle. In order to allow for different scenarios and to keep the model complexity manageable, we chose to parameterize the coupling functions as a mixture of two weighted two-dimensional Gaussians: $F_1 = K_1\, G_1(\theta, \phi) + K_2\, G_2(\theta, \phi)$ where $K_1$ and $K_2$ are coupling constants that can be positive or negative, and $G_i$ are Gaussians with arbitrary means and diagonal (but not necessarily isotropic) covariances (Supplementary Fig S6). To calibrate this model from the measured time traces, we factorized the probability of a sequence of measured peaks and divisions into a product of conditional probabilities that can be estimated numerically. We then computed the likelihood for entire datasets and optimized the

parameters using a genetic algorithm (details in Supplementary Information, Section II).

**Supplementary information** for this article is available online: http://msb.embopress.org

## Acknowledgements

We thank Emi Nagoshi for providing the NIH3T3-Rev-VNP-1 cells, the EPFL imaging facility (BIOP) for assistance with the microscopy, Mara de Matos for help with the shRNA cell lines, and Johannes Becker for useful discussions. The Naef Lab was supported by the Swiss National Science Foundation (SNF Grants 31-130714 and 31-153340), the European Research Council (ERC-2010-StG-260667), and the École Polytechnique Fédérale de Lausanne (EPFL). The laboratories of FN and DG received funding by the Fondation Leenaards and StoNets, a grant from the Swiss SystemsX.ch (www.systemsx.ch) initiative evaluated by the Swiss National Science Foundation (SNSF). DG acknowledges funding by SNSF professorship Grant PP00P3_128399.

## Author contributions

JB, RC, and FN designed and participated in the study concept. JB and RC acquired the data. JB, RC, KG, CG and FN analyzed and interpreted the data. JB, KG and FN wrote the manuscript. FN obtained the funding. All authors contributed to the preparation of the manuscript.

## Conflict of interest

The authors declare that they have no conflict of interest.

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
