## [Review Process File · Molecular Systems Biology]

Robust synchronization of coupled circadian and cell cycle oscillators in single mammalian cells

Jonathan Bieler, Rosamaria Cannavo, Kyle Gustafson, Cedric Gobet, David Gatfield, Felix Naef

Corresponding author: Felix Naef, Ecole Polytechnique Federale de Lausanne EPFL

Review timeline:	Submission date:	20 February 2014
	Editorial Decision:	20 March 2014
	Revision received:	14 May 2014
	Editorial Decision:	04 June 2014
	Revision received:	05 June 2014
	Accepted:	05 June 2014

Editor: Maria Polychronidou

Transaction Report:

1st Editorial Decision

20 March 2014

Thank you again for submitting your work to Molecular Systems Biology. We have now heard back from the three referees who agreed to evaluate your manuscript. As you will see from the reports below, the referees acknowledge that you analyze a potentially interesting topic. However, they raise a series of concerns, which should be carefully addressed in a revision of the manuscript.

Without repeating all the points listed below, one of the more fundamental issues refers to the need to include additional analyses (i.e. involving genetic and/or pharmacological perturbations) in order to more convincingly support the effect of the cell cycle on circadian period and the absence of control of the cell cycle by the circadian cycle.

On a more editorial level, we would like to mention that while we generally encourage the submission of individual Supplementary Figure/Table files, in exceptional cases (and depending on the nature of information provided) we allow the use of a single PDF file including a Table of Contents. As such, if you prefer to present the Supplementary Information in the single PDF file format, it would be fine.

If you feel you can satisfactorily deal with these points and those listed by the referees, you may wish to submit a revised version of your manuscript. Please attach a covering letter giving details of the way in which you have handled each of the points raised by the referees. A revised manuscript will be once again subject to review and you probably understand that we can give you no guarantee at this stage that the eventual outcome will be favorable.

Reviewer #1:

Previous studies have found that cell division is gated by the circadian clock. However, only one study demonstrated this directly by monitoring cell division and circadian cycling simultaneously and longitudinally in single mammalian cells (Nagoshi, 2004), and that one study did not include enough cells to be definitive. In the present study, the authors monitor circadian cycling in NIH3T3 fibroblasts using a RevErba-YFP fluorescent reporter, imaging over 8000 cells through 15,000 circadian peaks and 10,000 cell divisions. They find that cell divisions are clustered tightly ~5h before the circadian RevErba peak. This result is robust to manipulations of either the cell cycle (serum concentration, temperature) or circadian period (Cry2 knockdown, longdaysin). Furthermore, circadian period (or estimated instantaneous phase progression) is influenced by cell division in a circadian phase-dependent manner, but there is little evidence that cell cycle length is influenced by circadian phase.

Major Concerns:

A) The authors show that the circadian phase of cell division does not greatly affect the length of the cell cycle (Fig. S4B), and that their model is consistent with a predominant effect of the cell cycle on the circadian cycle rather than vice versa. However, given the strong clustering of cell divisions at a single circadian phase (Fig. 1C), it is misleading to suggest that cell division is not gated by the circadian clock in these cells.

B) As an alternative analytic method to determine causality in complex oscillatory systems, the authors may wish to consider the approach suggested by Sugihara et al. (Science 338:496, 2012).

C) More specific genetic or pharmacological manipulations of the cell cycle would strengthen the case for effects of the cell cycle on circadian period.

Minor Comments:

1) CT6 is mid-day, not morning (p. 5).

2) "Circadian phase progression" can be faster or slower, but "circadian phase" can only be earlier or later.

3) In Fig. 4B, explain that histograms are depicted.

4) Change "has lead" to "has led" (p. 3), "regulators have been shown... including CRY proteins" to "regulators including CRY proteins have been shown..." (p. 4), "theses" to "these" (p. 8), "Hela" to "HeLa" (p. 11), "in function" to "as a function" (p. 11), "the end the low" to "the end of the low" (p. 12), "cells types" to "cell types" (p. 14), "cell division," to "cell division." (p. 14), "par" to "per" (p. 11), "periods... was" to "periods... were" (p. 25).

Reviewer #2:

In this manuscript, Bieler et al. investigate the coupling between the cell cycle and the circadian clock in cultured NIH3T3 cells. This has been a subject of intense interest in recent years; one major contribution of this study is the application of a fluorescent imaging platform that allows them to track the circadian phase of a population of unsynchronized cultured fibroblasts using a RevErb-YFP reporter. With this system, they were able to convincingly demonstrate that cell divisions tend to occur about 5 hours prior to the peak of RevErb-YFP fluorescence (i.e. at roughly CT 1). In contrast to the case in other mammalian systems, NIH3T3 cells appear to show a predominant influence of the cell cycle on the circadian clock, not vice versa.

The authors also present results in which the circadian period has been lengthened, either by targeting Cry2 with an shRNA or by treatment with longdaysin. In both cases, the circadian period increases but synchronization between the cell cycle and the circadian clock is maintained.

Pg. 10: "...increasing the mean interval from divisions to the next peak." It isn't intuitively obvious (to me, at least) that a longer circadian period could result in a longer (d,p) interval -- if the 5-hour interval between the cell division and the circadian peak is controlled by the timing of the cell division, I would expect the interval between the preceding circadian peak and the cell division to get longer. Figure S7 isn't enough to convince me that there's a really strong effect in either direction, and I'm still a little confused by the fact that the cell cycle and circadian periods are clearly different but they still remain coupled. Is there a change in the number of division-free circadian periods?

I was a little disappointed not to see any selective perturbation of the cell cycle -- are there agents that can affect the period of the cell cycle in dividing cells without affecting the circadian clock (at least in non-proliferative cells)? The case for a causal cell cycle  clock relationship would be strengthened by selective perturbation of the cell cycle. The authors should either present data showing the effects of a perturbation of the cell cycle, or explain why this would not be feasible.

This paper was a pleasure to read; the writing is clear, the figures are easy to understand, and the arguments made are well-supported by the data presented. The primary audience is clearly circadian researchers. Some of the quantitative methods used, particularly for the fitting of an SDE model to noisy data extracted from fluorescence images, are likely to be of interest to quantitative biologists in other fields as well. I recommend its publication, once a few minor points have been addressed.

Minor points:

Fig. S1: It would be nice if the inset legend had units (i.e. 2-5% serum, etc.). Same story for Figure 4A,C,D

I'm not sure I completely understood the point that was being made by Figure S4 -- a little more explanation about why one would expect a positive or negative slope under different scenarios would help.

Pg. 6 (and figure S2 caption): Distributions are often described here as "more peaked" or "less peaked" -- including standard deviations (or even an appropriate statistical test) would make these comparisons more quantitative.

Pg. 7: "estimated the instantaneous circadian phase from the Rev-Erb -YFP signal" -- should be "from"

Figure 2D: Maybe it would be better to show the slope of the three traces (d_{θ}/d_t) instead of plotting $\theta(t)$. It's hard to see by eye whether the black curve is straight and the blue curve is inflected upward, or whether the black curve is inflected downward. It might even be better to plot d_{θ}/d_t against θ instead of t so that you can convert your horizontal pink and blue bars into vertical ones. Same comment for Figure S8.

Pg. 8: "Probabilities of theses" should be "probabilities of these"?

Pg. 8: "generic forms for the coupling function" -- other readers might feel differently about this, but I would have preferred to see the stochastic differential equations and the functional form of the coupling mentioned here, instead of having to wait until I got to the Materials and Methods section. I think this would have made the blue and yellow ellipses in Figs. 3-5 a lot easier to understand; I was a little confused about what they meant until I got to M&M and understood what the form of your coupling equation was.

Pg. 11: "Figure5A" would benefit from a space.

Pg. 11: "...in the dynamic." Did you mean "dynamics?"

Reviewer #3:

The study by Bieler et al. combines experiment and theory to investigate the mutual interaction of two fundamental oscillators in mammalian cells, i.e. the circadian oscillator and the cell cycle oscillator. The aim of the study is to characterize a potential synchronization of these oscillators and study the relative influence/dominance of one or the other. Experimentally they build on work already published 10 years ago (Nagoshi et al., Cell 2004), in which one of the major conclusions here is already presented, yet with a much lower number of cells and therefore less convincing statistics: i.e. cell division has an impact on circadian period.

Despite of this minor lack of novelty, overall this study is beautiful: the experiments are well

performed, the data vigorously analyzed and largely well presented. However, there are still major issues that need to be solved.

1. I am not yet convinced that there is really a very stable synchronization between the oscillators, because the phase relation (although being about -5 external hours) seems to be not stable. Actually the authors should not analyze phase relation in absolute, external hours but in circadian hours; then it would be apparent that the longer the peak-to-peak distance the later the cell division in the circadian cycle. This should be analyzed and discussed. It begs the question about cycle-to-cycle variation and potential "memory" effects. E.g. for the cases with long circadian cycles that peak early after YFP-peak (bottom traces in Fig. 1B), the next division seems to occur later than - 5hours, but maybe earlier than without a circadian oscillator.

2. The potential influence of the circadian oscillator is not vigorously tested in experiments. The period effect of Cry2 knockdown is way too subtle to allow clear analysis (the authors could have analyzed this at 40 {degree sign}C to increase the period mismatch), and longdaysin is inappropriate because of its effect on both oscillators. The authors should try another clock perturbation with a stronger phenotype (e.g. Fbx13 knockdown, Zhang et al., Maier et al.).

3. Presentation is often not intuitive. The figures should be labelled for easy understanding of both experimentalists and theoreticians. Here are some suggestions:

Fig. 1A: It would be nice to see time series of cells, with labelled ROI (which nucleus is followed for the traces shown?)

Fig. 2B/D: Use CT for the phase

Fig. 2D: Label colored region. Btw, I don't see any twisting of the blue line. Maybe an analysis of the slope at different time windows (as an inset figure) or a visual comparison with a straight line would help to see where phase progression is affected.

Fig. 3A/B: Label with "data" and "model"

Fig. 3C-F, 4E,F: Difficult to intuitively understand. Maybe label axes with "circadian time" and "cell cycle progression", mark the cell division and explain colors. Also in 3C, increase the size of the dots.

Fig 4A: add {degree sign}C to the temperatures

Fig. 4B: Provide the absolute number of traces analyzed for each condition (I assume the distributions are normalized).

34d etc. should read 34 {degree sign}C with division

Other comments/questions:

1. Fig. 4: Is the time of cell division really insensitive to period mismatch as the way of presenting the data suggests? Again (see point 1 above) analyze it with respect to CT. Oscillator theory predicts that phase relation depends on period mismatch (see e.g. Granada et al., Plos One 2013).

2. It is very interesting that temperature compensation seems not (or less) effective in dividing cells. The authors should analyze the Q10 and discuss this phenomenon.

3. Are the cells really unsynchronized - show a phase distribution as supplemental figure.

4. In the text: Try to avoid terms like early or late in the circadian cycle, since you relate this to Rev-Erb promoter activity, which is arbitrary. Better use CT times throughout the manuscript as reference (as already done at some occasions).

5. Is the model robust - what about the second and third best solutions? Are the results similar?

6. P.9, line 12 from below should read "Figure 4C"

Reviewer #1:

MSB-14-5218: Bieler et al., Mol Sys Biol

Previous studies have found that cell division is gated by the circadian clock. However, only one study demonstrated this directly by monitoring cell division and circadian cycling simultaneously and longitudinally in single mammalian cells (Nagoshi, 2004), and that one study did not include enough cells to be definitive. In the present study, the authors monitor circadian cycling in NIH3T3 fibroblasts using a RevErba-YFP fluorescent reporter, imaging over 8000 cells through 15,000 circadian peaks and 10,000 cell divisions. They find that cell divisions are clustered tightly ~5h before the circadian RevErba peak. This result is robust to manipulations of either the cell cycle (serum concentration, temperature) or circadian period (Cry2 knockdown, longdaysin). Furthermore, circadian period (or estimated instantaneous phase progression) is influenced by cell division in a circadian phase-dependent manner, but there is little evidence that cell cycle length is influenced by circadian phase.

Major Concerns:

A) The authors show that the circadian phase of cell division does not greatly affect the length of the cell cycle (Fig. S4B), and that their model is consistent with a predominant effect of the cell cycle on the circadian cycle rather than vice versa. However, given the strong clustering of cell divisions at a single circadian phase (Fig. 1C), it is misleading to suggest that cell division is not gated by the circadian clock in these cells.

We understand that we did not elaborate sufficiently on this important point (p. 7). In summary, our data had allowed us to conclude that the influence of the cell cycle on the circadian cycle was absolutely unambiguous. At the same time, such unidirectional influence turned out to be sufficient to quantitatively account for the synchronization of the two cycles. Furthermore, it was hard to imagine how a potential coupling from the circadian clock to the cell cycle would generate some of our observations, such as the shortening of the circadian interval in dividing cells.

We agree with the referee that distinguishing between the directionalities of the coupling using data from unperturbed conditions (in which the two cycles are synchronized) is difficult. Indeed, this is why we have collected enough single-cell data to have good statistics backed up by quantitative models. However, it is important to note that even in the unperturbed cells, fluctuations provided sufficient natural variation to reveal clear signatures of reverse coupling, namely: circadian intervals without divisions that were longer than those with divisions (Fig. 2A); and the varying length of the circadian interval with respect to the circadian phase at division (Fig. 2B). Similar signatures indicative of cell cycle gating by the circadian clock were not found. We had provided several arguments as to why this might be (higher variability of the cell-cycle, gating masked by the much stronger opposite coupling) in the Discussion (p. 16). We used modeling to investigate whether gating could have escaped detection in the simple statistical approach, but again, the models clearly showed that the strongest and most

consistent interaction was that of the cell cycle on the circadian cycle. Finally, the shCry2 experiments did not reveal differences in the cell cycle, which is now shown more clearly in our modified Fig. S10 (previously S7) (cf. our reply to reviewer 2, p. 5). We have done our best to avoid any overstatement regarding the exclusion of a potential gating of the cell cycle by the circadian clock in this system.

To strengthen these important points, we have now added additional independent experiments and analyses to the revised version of the manuscript. First, we used circadian-phase resetting experiments (new Fig. 7) to transiently perturb the circadian rhythm but not the cell cycle. As opposed to gene knock-downs, which may produce larger and/or unspecific side effects that can complicate interpretations, the minimally invasive perturbation of the circadian cycle (by dexamethasone or forskolin exposure) strongly indicated that circadian resetting did not influence cell-cycle progression. However, cell division strongly affected the circadian cycle (cf. our reply to reviewer 3, p. 10-11). Following on comment B below, we tested for causality using the Granger-Wald test, which again confirmed that evidence for gating of the cell cycle by the circadian clock was marginal compared to the much more reliable reverse influence of the cell-cycle on the circadian cycle (new Fig. S8 and Results p. 9, last paragraph).

We note that gating of cell division by the circadian cycle in mouse cells was described in the liver (Matsuo et al, 2003) and in primary fibroblasts (Kowalska et al, 2013), albeit not on the individual cell level. Now, taking our additional evidence into account, we conclude that circadian gating of the cell-cycle in NIH3T3 cells is likely to play a minor role, if any at all, in the synchronized dynamics of this coupled circadian and cell-cycle system.

B) As an alternative analytic method to determine causality in complex oscillatory systems, the authors may wish to consider the approach suggested by s et al. (Science 338:496, 2012).

We thank this reviewer for pointing us toward more general arguments to treat causality in time series data. After studying the Sugihara paper carefully, we concluded that the method is not directly applicable to our problem. As stated by the authors, the method is specifically aimed at a class of systems not covered by Granger Causality (GC). Indeed, while GC is adequate for strongly coupled and stochastic systems like ours, the Sugihara method is designed for weakly coupled and deterministic systems. However, this excellent reference prompted us to look into the classical Granger causality argument. In order to apply the Granger method, we needed to obtain temporal information on the cell-cycle progression. For this we took advantage of the under-used property that the size of the nucleus carries information on the cell-cycle progression (Fidorra et al., 1981). First, we noticed that the sizes of our 3T3 nuclei exhibit very similar temporal dependency as reported in Fidorra et al., 1981. We also verified that in HeLa cells transformed with the Fucci system, nuclear size can accurately predict G1/S transitions (new section “Nuclear area and cell cycle phase” in the Supplementary Information). Nuclear size cannot capture the details of cell-cycle transitions, but it does provide a proxy variable that tracks cell-cycle progression. The result of the Granger-Wald tests clearly indicates that

the cell-cycle Granger-causes the circadian phase progression much more consistently (in 55% of cells) than the opposite (12%). Thus the Granger method provides an independent argument that strengthens and supports our previous conclusions based on the modeling. These results are shown and discussed in the new Fig. S8 and in the last paragraph of p. 9 in the Results.

We are grateful to this reviewer for suggesting this method that prompted a more sophisticated causality analysis to support our initially suggestive data on the coupling direction.

C) More specific genetic or pharmacological manipulations of the cell cycle would strengthen the case for effects of the cell cycle on circadian period.

This is a very good suggestion and we have now performed additional recordings using both inhibitors of CDK2 (G1/S transition) and CDK1 (G2/M) kinases in a dose-dependent manner. The results clearly indicate that progressive blocking of cell-cycle progression with either inhibitor lengthens circadian intervals towards values that are statistically indistinguishable from those in non-dividing cells (see also our reply to a similar question by referee 2, p. 6 bottom). Thus, these additional experiments provide direct evidence that cell cycle progression causes shortening of circadian intervals. These results also substantiate our earlier conclusions from the temperature experiments (used to alter cell cycle duration without affect the circadian period) in which (p,d,p) intervals shortened at 40°C and lengthened at 34°C, compared to 37°C. In fact, it is interesting to note that the (p,p) and (p,d,p) intervals in the condition of 10 μ M CDK2 inhibitor at 37°C mimic those in the normal 34°C condition (Fig. 4B). Since these are important new experiments, we have added a new main Figure 5 and Results section (“Inhibition of the cell cycle lengthens circadian intervals and delays division phase”, p. 10-11) where we also highlight a phase delay in the divisions under both inhibitors that is fully consistent with generic properties of entrained oscillators, since the period mismatch between the driving cell cycle (the cell cycle is slowed down) and the circadian cycle is reduced under inhibition.

Minor Comments:

1) CT6 is mid-day, not morning (p. 5).

Thank you for pointing this out.

2) "Circadian phase progression" can be faster or slower, but "circadian phase" can only be earlier or later.

Thank you, this is absolutely correct, we have fixed this.

3) In Fig. 4B, explain that histograms are depicted.

We made this clear in the caption.

4) Change "has lead" to "has led" (p. 3), "regulators have been shown... including CRY proteins" to "regulators including CRY proteins have been shown..." (p. 4), "theses" to "these" (p. 8), "Hela" to "HeLa" (p. 11), "in function" to "as a function" (p. 11), "the end the low" to "the end of the low" (p. 12), "cells types" to "cell types" (p. 14), "cell division," to "cell division." (p. 14), "par" to "per" (p. 11), "periods... was" to "periods... were" (p. 25).

We appreciate the careful reading of our manuscript and apologize for these inaccuracies, which have been corrected.

Reviewer #2:

In this manuscript, Bieler et al. investigate the coupling between the cell cycle and the circadian clock in cultured NIH3T3 cells. This has been a subject of intense interest in recent years; one major contribution of this study is the application of a fluorescent imaging platform that allows them to track the circadian phase of a population of unsynchronized cultured fibroblasts using a RevErb-YFP reporter. With this system, they were able to convincingly demonstrate that cell divisions tend to occur about 5 hours prior to the peak of RevErb-YFP fluorescence (i.e. at roughly CT 1). In contrast to the case in other mammalian systems, NIH3T3 cells appear to show a predominant influence of the cell cycle on the circadian clock, not vice versa.

The authors also present results in which the circadian period has been lengthened, either by targeting Cry2 with an shRNA or by treatment with longdaysin. In both cases, the circadian period increases but synchronization between the cell cycle and the circadian clock is maintained.

Pg. 10: "...increasing the mean interval from divisions to the next peak." It isn't intuitively obvious (to me, at least) that a longer circadian period could result in a longer (d,p) interval -- if the 5-hour interval between the cell division and the circadian peak is controlled by the timing of the cell division, I would expect the interval between the preceding circadian peak and the cell division to get longer.

We agree that clarification is required here. First, thanks to both reviewers 2 and 3, we realized that we failed to clearly present the Cry2 data. Before discussing the phase shifting, we should have stressed that – consistent with the absence of cell cycle gating by the circadian cycle – the lengthening of the circadian cycle did not alter the mean cell-cycle duration. Indeed, the circadian period in shCry2 cells was 2.6 hours longer ($p < 10^{-16}$ compared to 37° C control), but the cell cycle duration was not changed. We have now rewritten the section on Cry2 and provided a much clearer Figure S10. Of note, we have also adopted here (Figure S10D) the excellent suggestion of reviewer 3 to show the distribution of circadian phases at divisions instead of the (p-d) intervals.

Coming to the reviewer's point, our argument was that in a scenario of a deterministic pulse-driven phase oscillator (the cell cycle shifting the circadian cycle at mitosis), which we take as the simplest conceptual model, an increased period mismatch results in a phase-advanced and less stable fixed point. However, due to the relatively small period lengthening, we only observed a modest but significant increase in the number of phase-advanced divisions, consistent with the above conceptual model. In addition, we added the relevant conclusion that the mechanism involved in mediating the coupling from the cell cycle to the circadian cycle likely does not involve CRY2. Thanks to these comments, we believe that our revised paragraph on Cry2 (Results, p. 11) is now much improved.

Figure S7 isn't enough to convince me that there's a really strong effect in either direction, and I'm still a little confused by the fact that the cell cycle and circadian periods are clearly different but they still remain coupled.

We have addressed this partially in the previous point. Note that in Figure S10E, we now clearly state that circadian intervals with divisions are shorter than without divisions in both conditions, which is a clear directional effect. We think there was a confusion related to the presentation of the shCry2 data. Let us clarify that Panel S10A shows the ‘free’ circadian intervals (p-p intervals, mean of 24 in wild-type and 25.5 in shCry2), while S10B shows the cell cycle duration (d-d intervals). Note that in the control cells, we found that circadian and cell-cycle intervals are highly correlated (Figure S3C, previously 1D). Here, while the (p,d,p) intervals are much shorter in shCry2, the fact that both (p,p) and (p,d,p) are longer than in wild-type can be explained by reduced synchrony caused by increased period mismatch.

Is there a change in the number of division-free circadian periods?

We used a strict protocol to control as best as we could for the proportion of dividing cells. Owing to confluency and other causes that affect the proportion of cells that have exited the cell cycle, this proportion is nevertheless subject to some variation (within a factor of maximally 2 across experiments, albeit usually less). In the manuscript (see page 5), there is already a mention of this in the context of the serum experiments where we state that “while serum concentration affected the fraction of mitotic cells, it had only a small effect on cell cycle duration (defined as the intervals between successive mitoses), and it showed no effect on the circadian”, and that this had negligible influence on the synchronization properties. In the case of shCry2 experiments we had less (40% reduction) division free intervals that in the controls.

I was a little disappointed not to see any selective perturbation of the cell cycle -- are there agents that can affect the period of the cell cycle in dividing cells without affecting the circadian clock (at least in non-proliferative cells)? The case for a causal cell cycle -- > clock relationship would be strengthened by selective perturbation of the cell cycle. The authors should either present data showing the effects of a perturbation of the cell cycle, or explain why this would not be feasible.

Thank you for this important suggestion (cf. our reply to referee 1, point C). These experiments are clearly feasible though somewhat delicate, since cell-cycle inhibitors tend to be quite toxic, which can be problematic given that we still need to record healthy cells for many days. Moreover, their dosage can be tricky as they tend to work in a “binary” (rather than gradual) manner such that the cell-cycle is mainly normal below a threshold concentration and is completely blocked above. Nevertheless, we identified both CDK2 (G1/S kinase) and CDK1 (G2/M) inhibitors (that are generally not altering division free (p-p) intervals) and succeeded in measuring their dose-dependent effects on (p-d-p) intervals. Our results nicely confirm the model that progressive lengthening of the cell cycle reverts the interrupted circadian intervals from shorter values to values comparable to (p-p) intervals. Note that the slight increase in both (p,p) and (p,d,p) intervals at high concentration (10 μ M) of CDK1 inhibitor may reflect that cells are then G2-arrested. It is conceivable that under such conditions the overall increase in transcription rates owing to doubled DNA content would contribute to period lengthening

as well, somewhat reminiscent of Dibner et al. 2007. Note also that the cells subjected to the CDK2 inhibitor at 10 μ M behave very similar to the cells recorded at 34°C, which supports our initial idea that reduced temperature affects circadian intervals through the cell cycle coupling. Another clear finding was that a lengthened cell cycle caused a delay in the division phase (the circadian phase at division), which is consistent with the same phase shifting argument as in the Cry2 case, except that now the phase is delayed since the period mismatch is reduced. We have thus included a new section in the Results on pages 10-11 “Inhibition of the cell cycle lengthens circadian intervals and delays division phase”, together with a new main Figure 5.

This paper was a pleasure to read; the writing is clear, the figures are easy to understand, and the arguments made are well-supported by the data presented. The primary audience is clearly circadian researchers. Some of the quantitative methods used, particularly for the fitting of an SDE model to noisy data extracted from fluorescence images, are likely to be of interest to quantitative biologists in other fields as well. I recommend its publication, once a few minor points have been addressed.

We greatly appreciate these positive comments.

Minor points:

Fig. S1: It would be nice if the inset legend had units (i.e. 2-5% serum, etc.). Same story for Figure 4A,C,D

Thanks for pointing this out and it has been revised. Note that Figure S1 is now Figure S2.

I'm not sure I completely understood the point that was being made by Figure S4 -- a little more explanation about why one would expect a positive or negative slope under different scenarios would help.

Thank you for pointing this out, we agree that this was not very clear. We have now explained this more clearly on page 7, bottom, and in the figure caption. The main point is that Figure S5B (formerly Figure S4B) can be explained entirely by the coupling from the cell-cycle onto the circadian cycle. While this argument was only suggestive of the absence of gating, we have, at the request of referee 1, significantly strengthened the causality analysis by applying a Granger-Wald test (new Figure S8, Results at bottom of p. 9, and Supplementary Information).

Pg. 6 (and figure S2 caption): Distributions are often described here as "more peaked" or "less peaked" -- including standard deviations (or even an appropriate statistical test) would make these comparisons more quantitative.

Thank you for pointing this out, we have avoided using this language and added statistical tests to compare distributions when appropriate in figure captions.

Pg. 7: "estimated the instantaneous circadian phase form the Rev-Erb α -YFP signal" -- should be "from"

Thanks for pointing out this typo.

Figure 2D: Maybe it would be better to show the slope of the three traces (d_{θ}/d_t) instead of plotting $\theta(t)$. It's hard to see by eye whether the black curve is straight and the blue curve is inflected upward, or whether the black curve is inflected downward. It might even be better to plot d_{θ}/d_t against θ instead of t so that you can convert your horizontal pink and blue bars into vertical ones. Same comment for Figure S8.

Thanks for this excellent suggestion that we have adopted in both figures. Note that while the analysis shows an overall slight increase in phase velocity (which probably reflects the limitations of the phase inference method), we do observe the previously described phase acceleration around mitosis. Note that Figure S8 is now Figure S11.

Pg. 8: "Probabilities of theses" should be "probabilities of these"?

Thank you.

Pg. 8: "generic forms for the coupling function" -- other readers might feel differently about this, but I would have preferred to see the stochastic differential equations and the functional form of the coupling mentioned here, instead of having to wait until I got to the Materials and Methods section. I think this would have made the blue and yellow ellipses in Figs. 3-5 a lot easier to understand; I was a little confused about what they meant until I got to M&M and understood what the form of your coupling equation was.

Thank you for this suggestion. While we felt that inserting a detailed model description here might disrupt flow of the text, we did add a minimal amount of notation and necessary definitions to facilitate understanding.

Pg. 11: "Figure5A" would benefit from a space.

Thank you.

Pg. 11: "...in the dynamic." Did you mean "dynamics?"

Thanks.

Reviewer #3:

The study by Bieler et al. combines experiment and theory to investigate the mutual interaction of two fundamental oscillators in mammalian cells, i.e. the circadian oscillator and the cell cycle oscillator. The aim of the study is to characterize a potential synchronization of these oscillators and study the relative influence/dominance of one or the other. Experimentally they build on work already published 10 years ago (Nagoshi et al., Cell 2004), in which one of the major conclusions here is already presented, yet with a much lower number of cells and therefore less convincing statistics: i.e. cell division has an impact on circadian period. Despite of this minor lack of novelty, overall this study is beautiful: the experiments are well performed, the data vigorously analyzed and largely well presented.

We thank the reviewer for this thoughtful assessment and are very pleased that he or she liked our work.

However, there are still major issues that need to be solved.

1. I am not yet convinced that there is really a very stable synchronization between the oscillators, because the phase relation (although being about -5 external hours) seems to be not stable. Actually the authors should not analyze phase relation in absolute, external hours but in circadian hours; then it would be apparent that the longer the peak-to-peak distance the later the cell division in the circadian cycle. This should be analyzed and discussed.

We are very grateful for this excellent point. We have now systematically added the analysis in terms of circadian phase (Figures 1, 3, 5, S10), which indeed makes the story even more interesting. We have also kept the distributions in absolute time, since their tightness likely conveys a signature of the mechanism, as emphasized in the Discussion, and we find it interesting to contrast the two representations. There are two aspects to the reviewer's question. First, a 1:1 synchronization in a stochastic setting implies that the distributions of division phases are peaked and unimodal, which we observe everywhere; and secondly, as the referee mentions below, the most likely phase depends on period mismatch. Thus, the peak in the division phases should shift in the temperature experiments (Figure 4), the Cry2 condition (Figure S10), and now in our newly added experiments on cell-cycle inhibition (Figure 5). It turns out that all these experiments support the prediction, in a scenario where the cell cycle unilaterally entrains the circadian cycle, that larger period mismatch (shCry2 or 40°C) results in phase-advanced divisions, while reduced period mismatch (CDK1/2 inhibitors) produces phase-delayed divisions. Since this is another strong argument in favor of a dominant effect of the cell cycle on the circadian cycle, we now discuss this more clearly in the Discussion (p. 16).

It begs the question about cycle-to-cycle variation and potential "memory" effects. E.g. for the cases with long circadian cycles that peak early after YFP-peak (bottom traces in

Fig. 1B), the next division seems to occur later than - 5hours, but maybe earlier than without a circadian oscillator.

This is a great idea. We have indeed tried to look into such effects, but did not find anything worth pursuing at this point. For example, we analyzed the correlation of (p1,d2) and (p3,d4) intervals across all (p1,d2,p3,d4). While we found a small positive correlation indicative of memory effect, the low statistics on the rare early divisions make us cautious, and we would probably have to perform further targeted experiments to look into this more seriously in the future.

2. The potential influence of the circadian oscillator is not vigorously tested in experiments. The period effect of Cry2 knockdown is way too subtle to allow clear analysis (the authors could have analyzed this at 40 {degree sign}C to increase the period mismatch), and longdaysin is inappropriate because of its effect on both oscillators. The authors should try another clock perturbation with a stronger phenotype (e.g. Fbx13 knockdown, Zhang et al., Maier et al.).

We thank the reviewer for such useful suggestions. As already mentioned in our responses to referee 2, we realize that we did not present the Cry2 data optimally and have now significantly re-written this paragraph (Results, p. 11) and shown a better quantification of the data (Fig. S10E). In our hands the shCry2 lengthens the circadian period by 2.6 hours, and we do not fully agree that the resulting data is uninformative: the answers eventually rely on statistical analysis. Since this point overlaps with referee 2, we refer to our reply to referee 2 on p. 5-6 for a detailed clarification on the shCry2 experiment.

Beyond the improvement in shCry2 analysis, we also followed the reviewer's suggestion and generated a 3T3-Venus shRNA cell line stably expressing an Fbx13-targeting shRNA. These cells showed a mean period lengthening of 2.2 hours. This is less than the most efficient siRNAs found for human cells in Maier et al., but comparable to the other constructs in that paper. We then performed microscopy experiments and obtained very similar results to the shCry2, with one minor difference. Cell-cycle duration (measured as intervals between successive divisions) was about one hour longer in Fbx13 shRNA cells (Figure R1, p.14 below). Since all other properties were again consistent with a dominant interaction from the cell to the circadian cycle, including the shortening of circadian intervals with divisions, we tend to believe that the slight lengthening of the cell cycle reflects a promiscuous role of FBXL3 in mitosis, since this protein interacts with the G1/S regulator CDC34 (<http://severus.dbmi.pitt.edu/wiki-pi/index.php/pair/view/997/262> 24, and Cenciarelli et al., Curr Biol 1999). Overall, we concluded that the Fbx13 experiment does not provide significant additional insights to justify inclusion as a new figure (since we now have 7 main figures and 12 supplementary figures already), although we would be happy to include it if the reviewer suggests (Figure R1, p.14).

As genetic manipulations always carry the risk of such unwanted secondary and side effects, we tried to address the reviewer's concern using a less invasive method to perturb the circadian clock and record a possible influence on the cell cycle. To this end, we

chose a transient (rather than permanent) perturbation to the circadian cycle in the form of dexamethasone- and forskolin- induced phase resetting (new Fig. 7 and Results section: “Circadian phase resetting does not influence cell divisions but transiently perturbs synchronization of circadian and cell cycles”, p. 13), which gave a beautiful confirmation of our previous results.

First, we found that circadian phase resetting did not influence the cell-division times, which are practically uniformly distributed in time in all conditions including controls. This indicated that circadian-to-cell-cycle gating was not operating in our cells. In addition, the data (shown as raster plots and quantified using synchronization indices) clearly show that as soon as divisions occur, the circadian peaks follow, while the circadian cycles in non-dividing cells remain aligned with the initial dexamethasone shock. Finally, synchronization of circadian and cell cycles, while decreased after the treatment, progressively relaxed to the levels of the untreated cells, as we would expect from a transient perturbation. Thus, we believe that these experiments unambiguously demonstrate that the circadian phase did not influence cell cycle progression, while the converse effect is very clear.

3. Presentation is often not intuitive. The figures should be labelled for easy understanding of both experimentalists and theoreticians.

Thanks for this feedback, we have taken care of this in all relevant figures.

Here are some suggestions:

Fig. 1A: It would be nice to see time series of cells, with labelled ROI (which nucleus is followed for the traces shown?)

We agree, and we have added a series of snapshots of two single-cell traces as supplemental Figure S1, showing clearly the tracked cells by their YFP signal and indicating the signatures of circadian peaking and cell division.

Fig. 2B/D: Use CT for the phase

We believe that circadian phase is better defined for stochastic oscillations because it refers to an internal state of the oscillator, and does not depend on period variability, or how fast a particular cell progresses through these states. For example CT23 does not exist for dividing cells with a period of 22 hours. Note that at the excellent request of reviewer 2, we have changed Fig. 2D.

Fig. 2D: Label colored region. Btw, I don't see any twisting of the blue line. Maybe an analysis of the slope at different time windows (as an inset figure) or a visual comparison with a straight line would help to see where phase progression is affected.

Thanks for this, we agree. The new version of this plot solves these issues (cf. our comment to referee 2 regarding the speed up of phase progression in late dividing cells, p. 8).

Fig. 3A/B: Label with "data" and "model"

Thank you for this suggestion.

Fig. 3C-F, 4E,F: Difficult to intuitively understand. Maybe label axes with "circadian time" and "cell cycle progression", mark the cell division and explain colors. Also in 3C, increase the size of the dots.

Indeed, we have now labeled these plots more intuitively.

Fig 4A: add {degree sign}C to the temperatures

Thanks for pointing out this omission.

Fig. 4B: Provide the absolute number of traces analyzed for each condition (I assume the distributions are normalized).

34d etc. should read 34{degree sign}C with division

We have added the number of traces and fixed the degree sign.

Other comments/questions:

1. Fig. 4: Is the time of cell division really insensitive to period mismatch as the way of presenting the data suggests?

Yes, the time (in hours) of cell division measured with respect to the next circadian peak is insensitive, but division phases depend on period mismatch (see point 1 above).

Again (see point 1 above) analyze it with respect to CT.

See our reply to point 1.

Oscillator theory predicts that phase relation depends on period mismatch (see e.g. Granada et al., Plos One 2013).

This is correct, we have addressed this already in point 1 and cited this reference in the text.

2. It is very interesting that temperature compensation seems not (or less) effective in dividing cells. The authors should analyze the Q10 and discuss this phenomenon.

Thank you very much for pointing this out. We have always thought of temperature compensation in the context of the unperturbed circadian cycle, but we agree that the idea of temperature compensation being dependent on cell proliferation status would clearly be interesting for chronobiologists. We have thus added a point on this (Results p. 10, top).

3. Are the cells really unsynchronized - show a phase distribution as supplemental figure.

The reviewer rightly points to the fact that it is notoriously difficult to obtain reliably unsynchronized circadian cycles since even splitting cells, for example, will tend to partially synchronize their oscillators. While we have applied a strict protocol where we plate cells one day before starting the three-day recordings, we indeed find (as now shown in the new Figures 7 and S13) that the circadian cycle in unstimulated cells is weakly synchronized (only slightly more than the cell cycle, $R \sim 0.2$ vs. $R \sim 0.15$), certainly much less than after dex stimulation. The corresponding phase distributions are now shown in Figure S13. However, this residual weak synchrony of circadian phases does not influence our analysis as long as the system is at steady state with respect to the relationship between circadian and cell cycle (see new Fig. 7).

4. In the text: Try to avoid terms like early or late in the circadian cycle, since you relate this to Rev-Erb promoter activity, which is arbitrary. Better use CT times throughout the manuscript as reference (as already done at some occasions).

We appreciate this comment, since we felt that early and late is the most intuitive in the context of the *Rev-Erba* peaks, we have used ‘early or late in the circadian interval’, but also referred to CT times on p 5, middle; p 6, bottom; p 15, top. If anything we should use normalized CT, which is equivalent to the phase (see our reply to point 3 on p. 11).

5. Is the model robust - what about the second and third best solutions? Are the results similar?

Yes, they are. We had already addressed this in Figs. 3E-F and 4E-F by showing 36, respectively 38, independent solutions. In addition, the numbers were provided in Tables S2-S5.

6. P.9, line 12 from below should read "Figure 4C"

We have fixed this.

Figure R1. Fbx13 deficient cells with longer circadian periods show reduced synchrony.

A. Circadian intervals (p1,p2) in shFbx13 cells are significantly longer than controls ($p < 2 \times 10^{-8}$ for the 37 °C dataset in solid black, t-test). Means are 25.9 +/- 6.3 (SD) in shFbx13 and 23.7 +/- 3.1 in the controls. B. The cell cycle duration is significantly longer in the shFbx13 cells ($p < 0.0036$, t-test). C. The intervals from divisions to the next circadian peaks (d,p) are slightly lengthened in the shFbx13 cell line ($p < 5 \times 10^{-6}$, Kolmogorov–Smirnov test, K-S). D. The circadian phases at division are slightly more peaked in shFbx13 cells compared to controls (K-S test ; $p < 5.9 \times 10^{-5}$). E. Mean circadian intervals with divisions are significantly shorter than intervals without divisions in both control ($p < 10^{-16}$, t-test) and shFbx13 cells ($p < 0.0027$, t-test). Mean cell cycle duration is shown in red. The error bars show the standard error on the mean. The total number of shFbx13 cell traces analyzed is $n=340$.

2nd Editorial Decision

04 June 2014

Thank you again for submitting your work to Molecular Systems Biology. We have now heard back from the two referees who were asked to evaluate your manuscript. As you will see below, the referees think that their main concerns have been satisfactorily addressed. Referee #2 lists a number of minor comments, which we would ask you to address in a revision of the manuscript.

On a more editorial level, we would like to draw your attention to the following:

- Please provide individual figure files for the *main figures*.
- An Author Contribution statement needs to be included.
- We would like to ask you to provide a "thumbnail image" (211x157 pixels, jpeg format) to highlight the paper on our homepage.

For more information regarding the points above please refer to our Author Guidelines <<http://msb.embopress.org/authorguide>>.

Please resubmit your revised manuscript online, with a covering letter listing amendments and responses to each point raised by the referees. Please resubmit the paper ****within one month**** and ideally as soon as possible. If we do not receive the revised manuscript within this time period, the file might be closed and any subsequent resubmission would be treated as a new manuscript. Please use the Manuscript Number (above) in all correspondence.

Reviewer #2:

In this revised manuscript, Bieler and co-workers have taken pains to clarify the presentation of their results. The clarity and focus of the presentation have improved substantially. I particularly appreciate the inclusion of experiments dealing with perturbations to the cell cycle length and with circadian phase shifting.

All of my major concerns have been addressed; I noticed a couple of minor typographical errors:

Abstract: "In principle, such synchrony may be caused by coupling from one oscillator onto the other, or in both directions. either direction, or both." -- cut and paste error?

Pg. 18: "CO₂" -- the "2" should be subscripted.

Supplementary Information, pg. 29, Figure M16 caption: I suspect you meant "nucleus", not "nuclus"

Reviewer #3:

The authors responded very well to all of my concerns

2nd Revision - authors' response

05 June 2014

Replies to Reviewer 2:

Abstract: "In principle, such synchrony may be caused by coupling from one oscillator onto the other, or in both directions. either direction, or both." -- cut and paste error?

Yes indeed, thanks so much for catching this. We have corrected the sentence.

Pg. 18: "CO₂" -- the "2" should be subscripted.

We have corrected this.

Supplementary Information, pg. 29, Figure M16 caption: I suspect you meant "nucleus", not "nuclus"

We have corrected this.